# MM-Fi: Multi-Modal Non-Intrusive 4D Human Dataset for Versatile Wireless Sensing

**Jianfei Yang**[1,*]**, He Huang**[1]**, Yunjiao Zhou**[1]**, Xinyan Chen**[1]**, Yuecong Xu**[1]**,
Shenghai Yuan**[1]**, Han Zou**[1]**, Chris Xiaoxuan Lu**[2]**, Lihua Xie**[1]
[1]School of Electrical and Electronic Engineering, Nanyang Technological University
[2]School of Informatics, University of Edinburgh
Project Page: https://ntu-aiot-lab.github.io/mm-fi
Dataset Toolbox: https://github.com/ybhbingo/MMFi_dataset

## Abstract

4D human perception plays an essential role in a myriad of applications, such as home automation and metaverse avatar simulation. However, existing solutions which mainly rely on cameras and wearable devices are either privacy intrusive or inconvenient to use. To address these issues, wireless sensing has emerged as a promising alternative, leveraging LiDAR, mmWave radar, and WiFi signals for device-free human sensing. In this paper, we propose MM-Fi, the first multi-modal non-intrusive 4D human dataset with 27 daily or rehabilitation action categories, to bridge the gap between wireless sensing and high-level human perception tasks. MM-Fi consists of over 320k synchronized frames of five modalities from 40 human subjects. Various annotations are provided to support potential sensing tasks, *e.g.*, human pose estimation and action recognition. Extensive experiments have been conducted to compare the sensing capacity of each or several modalities in terms of multiple tasks. We envision that MM-Fi can contribute to wireless sensing research with respect to action recognition, human pose estimation, multi-modal learning, cross-modal supervision, and interdisciplinary healthcare research.

## 1   Introduction

Human sensing and modeling serve as the fundamental technology in computer vision, human-computer interaction, ubiquitous computing, and computer graphics [16]. Accurately recognizing human activities and reconstructing human pose empower a broad spectrum of applications, *e.g.*, gaming, home automation, autonomous driving, augmented and virtual reality, animation, and rehabilitation. However, existing methods mainly rely on cameras [25] and wearable inertial sensors [50], where significant limitations exist when applied in real-world scenarios. For instance, cameras result in privacy concerns in domestic settings and are susceptible to lighting conditions. Wearable inertial sensors are low-cost solutions but require strong user compliance for wearing them at all time.

Recently, wireless human sensing has emerged as a promising solution that leverages non-intrusive sensors such as LiDAR, mmWave radar, and WiFi to address the limitations of illumination, privacy, and inconvenience [27]. These device-free sensors spread laser or radio frequency (RF) signals whose responses reflect human motions in different levels of granularity: high-resolution point cloud data from LiDAR [21], medium-resolution point cloud from mmWave radar [2], and low-resolution channel state information (CSI) from WiFi [47, 56]. These data modalities are complementary to existing camera-based or device-based solutions and enable more privacy-preserving human sensing applications such as homes and hospitals.

In this work, we present **MM-Fi**, a multi-modal non-intrusive 4D (spatial-temporal) human dataset for high-fidelity human sensing to facilitate the algorithm development of wireless human sensing.

---

[*]Corresponding authors (yang0478@e.ntu.edu.sg)

37th Conference on Neural Information Processing Systems (NeurIPS 2023) Track on Datasets and Benchmarks.

MM-Fi consists of 1080 consecutive sequences with over 320k synchronized frames from five sensing modalities: RGB image, depth image, LiDAR point cloud, mmWave radar point cloud, and WiFi CSI data. The dataset includes annotations for 2D/3D human pose landmarks, action categories, 3D human position, and estimated 3D dense pose. To the best of our knowledge, MM-Fi is the first dataset that comprises five non-intrusive modalities for 4D human pose estimation (HPE). The contributions and features of MM-Fi are listed below.

**Multiple Sensing Modalities.** MM-Fi contributes to multimodal human sensing by considering the potentially complementary nature of different sensing modalities. MM-Fi provides five non-intrusive sensing modalities including RGB frames, depth frames, LiDAR point cloud, mmWave radar point cloud, and WiFi CSI data, with kinds of annotations, *i.e.*, 2D pose landmarks of an RGB camera and stereo camera, 3D pose landmark, action category, and 3D human position.

**Synchronized Mobile Sensor Platform.** MM-Fi is collected by a novel customized platform that captures and synchronizes multiple sensors through a mini-PC running the Robot Operating System (ROS). The mobility of our platform enables us to collect data in diverse environments.

**Profuse Action Sets.** MM-Fi consists of 27 categories of human actions including 14 daily actions and 13 clinically-suggested rehabilitation actions. Therefore, MM-Fi can contribute to ubiquitous computing, *e.g.*, home automation applications, and healthcare research, *e.g.*, the evaluation and recovery of neuroscience disorders or physical body injury.

**Versatile Sensing with Unexplored Tasks.** The rich variety of data modalities and annotations in MM-Fi enables diverse sensing tasks, such as multi-modal fusion and cross-modal supervision for human sensing. Furthermore, MM-Fi opens up new research possibilities for previously unexplored tasks, *e.g.*, human pose estimation using the combinations of two or three sensor modalities, and unexplored problems, *e.g.*, domain generalization in wireless multi-modal sensing.

**Extensive Benchmarks.** To facilitate future research, we release the pre-processed dataset, various annotations, the codes for data loading, and extensive benchmarks on multi-modal human pose estimation and skeleton-based action recognition.

## 2 Related Work

### 2.1 Human Pose Estimation

3D human pose estimation has been extensively studied with the deployment of various sensing schemes including marker-based sensing and marker-less sensing, in which the marker-less sensing is more widely accepted for its non-intrusive and user-friendly properties [37, 40, 2, 3, 7, 14, 36, 46]. Vision-based approaches take over the majority of 3D HPE owing to the popularity of cameras, where the 2D keypoints are usually recognized from RGB frames and a deep learning model will be proposed to generate 3D joints based on a set of 2D keypoints [37]. As a result, RGB and depth cameras are also included in our dataset. Though RGB cameras produce high-resolution images, they suffer in poor lighting conditions and are fragile to weather conditions [7]. Depth cameras are capable of providing dense point clouds with the texture of human body, but they still face the problems of high noise and outliers [21]. For more stable and accurate sensing, we also consider exploring LiDAR for 3D HPE in this dataset, which is previously applied in industrial fields such as SLAM and autonomous driving [54]. Compared with cameras, LiDAR is more robust to noise and thus obtains more reliable human body texture [21]. Also, the cost of LiDAR devices is gradually falling as a result of the recent laser technology development. However, it would also be affected by occlusion, which makes the point cloud texture less representative. Despite the accuracy, LiDAR is still expensive for many scenarios and only realistic for a limited amount of applications. Besides vision-based sensing, radio frequency sensing is a promising technique for 3D HPE [52, 27]. mmWave-based HPE has been paid increasing attention thanks to the comparable performance to vision-based approaches and the privacy-concerned merit [36, 1]. Thus, mmWave is exploited in our dataset. In addition to mmWave-based sensing, WiFi CSI-based sensing is also emerging recently. WiFi sensing has been reported in several applications (*e.g.*, respiratory monitoring, gesture, and action recognition.) but is seldom seen for 3D HPE [58, 57, 46]. Now, our dataset includes WiFi sensing for related research. Furthermore, the obtained 3D human poses can be applied to realize action recognition tasks.

| Dataset | RGB | Depth | LiDAR | mmWave | WiFi | Action | 3DPOS | 2DKP | 3DKP | 3DDP | # Subj | # Act | # Seq | # Frame |
|---|---|---|---|---|---|---|---|---|---|---|---|---|---|---|
| COCO [23] | ✓ | - | - | - | - | - | - | ✓ | - | - | - | - | - | 104k |
| MPII [4] | ✓ | - | - | - | - | ✓ | - | ✓ | - | - | - | 410 | 24k | 25k |
| MPI-INF-3DHP [24] | ✓ | - | - | - | - | - | - | ✓ | ✓ | - | 8 | 8 | 16 | 1.3M |
| CMU Panoptic [18] | ✓ | ✓ | - | - | - | - | - | ✓ | ✓ | - | 8 | 5 | 65 | 154M |
| Human3.6M [16] | ✓ | ✓ | - | - | - | ✓ | - | ✓ | ✓ | - | 11 | 17 | 839 | 3.6M |
| NTU RGB+D [37] | ✓ | ✓ | - | - | - | ✓ | - | ✓ | ✓ | - | 40 | 60 | 56k | 4M |
| 3DPW [40] | ✓ | - | - | - | - | - | - | ✓ | ✓ | - | 7 | - | 60 | 51k |
| MPI08 [30] | ✓ | - | - | - | - | - | - | - | ✓ | - | 4 | 24 | 24 | 14k |
| TNT15 [41] | ✓ | - | - | - | - | ✓ | - | - | ✓ | - | 1 | 5 | - | 14k |
| MoVi [14] | ✓ | - | - | - | - | ✓ | - | ✓ | ✓ | - | 90 | 21 | 1044 | 712k |
| LiDARCap [21] * | ✓ | - | ✓ | - | - | ✓ | - | - | ✓ | - | 13 | 20 | - | 184k |
| LIPD [34] * | ✓ | - | ✓ | - | - | ✓ | - | - | - | - | 15 | 30 | - | 62.34k |
| LiCamPose [10] * | ✓ | - | ✓ | - | - | ✓ | - | - | ✓ | - | - | 6 | - | 8.98k |
| SLOPER4D [11] | ✓ | - | ✓ | - | - | - | - | - | - | - | 12 | - | 15 | 100k |
| CIMI4D [45] | ✓ | - | ✓ | - | - | - | - | ✓ | ✓ | - | 12 | - | 42 | 179.84k |
| Waymo [54] * | ✓ | ✓ | ✓ | - | - | - | - | ✓ | ✓ | - | 13 | - | 1950 | 184k |
| HuMMan [7] | ✓ | ✓ | ✓ | - | - | ✓ | - | ✓ | ✓ | - | 1000 | 500 | 400k | 60M |
| RF-Pose [52] * | ✓ | - | - | ✓ | - | ✓ | - | ✓ | - | - | 100 | 1 | - | - |
| RF-Pose3D [53] * | ✓ | - | - | ✓ | - | ✓ | - | ✓ | ✓ | - | >5 | 5 | - | - |
| mmPose [36] * | - | - | - | ✓ | - | ✓ | - | - | ✓ | - | 2 | 4 | - | 40k |
| mmMesh [44] * | ✓ | - | - | ✓ | - | ✓ | - | - | ✓ | - | 20 | 8 | - | 3k |
| MARS [2] | - | - | - | ✓ | - | ✓ | - | - | ✓ | - | 4 | 10 | 80 | 40k |
| mmBody [8] | ✓ | ✓ | - | ✓ | - | ✓ | - | - | ✓ | - | 20 | 100 | - | 200k |
| mRI [1] | ✓ | ✓ | - | ✓ | - | ✓ | - | ✓ | ✓ | - | 20 | 12 | 300 | 160k |
| AHA-3D [5] | ✓ | - | - | - | - | ✓ | - | - | ✓ | - | 21 | 4 | 79 | 170k |
| HPTE [6] | ✓ | ✓ | - | - | - | ✓ | - | - | ✓ | - | 5 | 8 | 240 | 100k |
| WiPose [17] * | ✓ | - | - | - | ✓ | ✓ | - | - | ✓ | - | 10 | 16 | - | 96k |
| GoPose [33] * | ✓ | - | - | - | ✓ | ✓ | - | - | ✓ | - | 10 | >9 | - | 676.2k |
| **MM-Fi** | ✓ | ✓ | ✓ | ✓ | ✓ | ✓ | ✓ | ✓ | ✓ | ✓ | 40 | 27 | 1080 | 320.76k |

Table 1: Comparisons of MM-Fi with published datasets. * denotes that the dataset is not accessible. The proposed MM-Fi includes all five non-intrusive sensors, and has four types of annotations including action category (Action), 3D subject position (3DPOS), 2D and 3D whole-body keypoints (2DKP and 3DKP), and 3D dense pose (3DDP). Compared to existing RGB-D datasets, MM-Fi has more sensor modalities and various annotations.

## 2.2 Multi-modal Human Dataset

With the emergence of applications to human pose estimation, multi-modal datasets with annotations will strongly support the research in relevant fields. Previously published datasets on HPE and our dataset are summarized and compared in Table 1. Traditional datasets on HPE mainly rely on RGB or depth frames, which provide 2D or 3D keypoints as the ground truth (*e.g.*, COCO [23], NTU RGB+D [37]). Recent works on radio frequency-based HPE apply mmWave (some assisted with RGB images) as the sensing modality (*e.g.*, mmPose [36], RF-Pose3D [53] with RGB). Waymo [54] and HuMMan [7] are the popular choices for LiDAR-based HPE algorithms development. In addition, with the increasing attention in WiFi sensing field, datasets with WiFi sensory inputs have been published for pose estimation with 3D annotations and action labels (*e.g.*, WiPose [17], GoPose [33]). Despite the development of the aforementioned works, there is still much room for HPE dataset. First, the mentioned datasets [23, 37, 36, 2, 17] support no more than three non-intrusive modalities, making it difficult to develop multi-modal sensing systems. Second, the quantity of most datasets related to mmWave and WiFi sensing is not sufficient in both the number of subjects and actions and the domain settings, which can impact the performance of wireless sensing. While recent works such as the mRI dataset [1] have made promising progress by supporting multiple modalities, including maker-based and maker-less sensing techniques (i.e., IMU, RGB, depth and mmWave), the aforementioned challenges still remain due to the lack of LiDAR and WiFi modalities and multiple domains. To address these challenges, our MM-Fi emphasizes the diversity of modalities and the variety of domain settings. Meanwhile, our dataset is equipped with richer data content in comparison to the majority of the above datasets (both in the number of subjects and actions). To the best of our knowledge, the MM-Fi is the first 3D HPE dataset with most of the non-intrusive sensing modalities, including RGB, depth, LiDAR, mmWave, and WiFi [15, 35], and with multiple kinds of annotations. MM-Fi thus has the potential to contribute to various research tasks in machine learning, computer vision, ubiquitous computing, and healthcare.

## 3 Sensor Platform

To facilitate data collection for our MM-Fi dataset, we develop a customized sensor platform that consists of the Intel RealSense D435 camera, the Texas Instrument IWR6843 60-64GHz mmWave

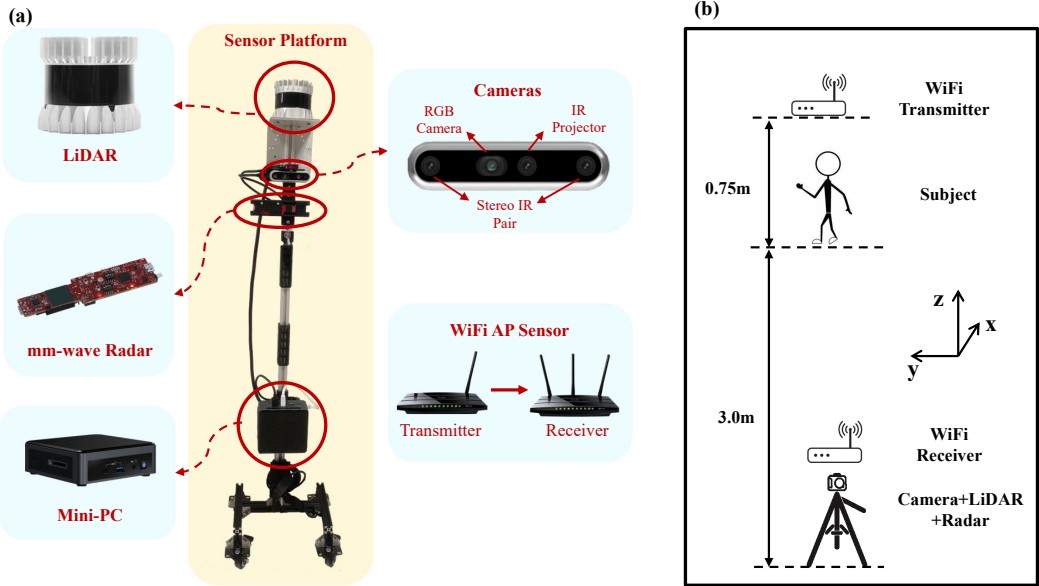

Figure 1: Overview of the experimental setup for data collection. (a) shows the customized sensor platform with one mini-PC for data synchronization and four sensor devices including Ouster OS1 32-channel LiDAR, TI IWR6843 mmWave radar, Intel Realsense D435 stereo depth camera, and TP-Link N750 WiFi APs. (b) shows the layout of data collection where all subjects are positioned 3.0m away from the platform. The WiFi transmitter is placed 0.75m away from the subject.

radar, the Ouster OS1 32-line LiDAR, and a pair of TP-Link N750 WiFi APs. As shown in Figure 1, all the sensors and one mini-PC are integrated into a mobile platform with ROS installed on the mini-PC for data collection and synchronization, with the platform placed 3.0 meters away from the subject during data collection. The features of these sensor modalities are compared in Table 2.

## 3.1 Sensor Modalities

**RGB-D Frames from Cameras.** We utilize an Intel Realsense D435 stereo depth camera to capture RGB-D frames. The camera consists of one high-precision RGB camera and two infra-red cameras with the resolutions of 1920×1080 and 1280×720, respectively. We calibrate the three cameras and consider the center of the RGB camera as the origin of the world coordinate system.

**Point Cloud from Radar and LiDAR.** Our point cloud data is obtained using a Texas Instruments (TI) IWR6843 mmWave radar and an Ouster OS1 32-channel LiDAR. The LiDAR point clouds are dense and cover the full range of human movement, with each LiDAR point $P_l$ represented by the spatial coordinates $P_l = (x, y, z)$. In contrast, the IWR6843 mmWave radar generates more sparse point clouds by emitting Frequency Modulated Continuous Waves (FMCW), which vary the frequency of the transmitted signal by a modulating signal at a known rate over a fixed time period. The reflected signals are used to measure the frequency difference and Doppler frequency for computing the distance and the speed of an object respectively. In this manner, a single point in a mmWave radar point cloud is represented as $P_m = (x, y, z, d, I)$, where $(x, y, z)$ indicates the spatial coordinates, $d$ denotes the Doppler velocity, and $I$ denotes the signal intensity. It is found that the number of points in one frame is too small for mmWave radar and sometimes the empty frame may appear due to the hardware instability. To address this issue, we aggregate each mmWave frame using the adjacent frames within its consecutive $0.5s$ period to increase the number to around 128. This strategy has been previously proposed in radar-based human pose estimation method [3].

**WiFi CSI Data.** WiFi CSI data describes the propagation link from the transmitter to the receiver. Recent studies have shown human movements can affect the CSI data extracted from WiFi signals, enabling WiFi-based human sensing which is ubiquitous, privacy-preserving, and cost-effective [46]. For WiFi CSI data collection, we built the WiFi sensing system [48] based on two Commercial Off-The-Shelf (COTS) WiFi Access Points, TP-Link N750, and the Atheros CSI Tool [43]. The platform

| Modality | Rate | Freq. | Privacy | Illum. | Range | Granularity | Cost | Data |
|---|---|---|---|---|---|---|---|---|
| RGB | 30Hz | - | * | * | ** | *** | ** | RGB frame |
| Depth | 30Hz | - | ** | ** | ** | ** | ** | Depth and infra-red frame |
| LiDAR | 20Hz | 360THz | *** | *** | *** | *** | *** | Point cloud |
| mmWave | 30Hz | 60-64GHz | *** | *** | *** | ** | ** | Sparse point cloud |
| WiFi | 1000Hz | 5GHz | *** | *** | * | * | * | CSI frame |

Table 2: Comparisons of five non-intrusive sensors. Rate: Sampling rate. Freq.: Signal frequency. Privacy: privacy-preserving ability. Illum.: Robustness to illumination. Range: Sensing range. Resolution: Data resolution. Cost: cost of sensors.

runs at 5GHz with a bandwidth of 40MHz, enabling the collection of CSI data with 114 subcarriers per pair of antennas at up to a sampling rate of 1000Hz. As shown in Figure 1, we embedded the receiver with three antennas on the sensor platform, and the transmitter with one antenna is placed on the other side. For the smoothness and stability of CSI data, we have implemented the average sliding window method inside the firmware of the sensing platform, which could produce a CSI stream of about 100Hz. Besides, due to the inconsistency of data acquisition rate between different modalities, the CSI data is further augmented to form a $3 \times 114 \times 10$ matrix within a time period of 100ms.

## 3.2 Synchronization

Synchronization is an indispensable prerequisite for a multi-modal human dataset. To achieve synchronization, we connect all the sensors and the WiFi receiver to the same mini-PC and develop a data collection system using the Robot Operating System (ROS) [31]. In ROS, all the sensor modalities are saved in a ROS bag with the timestamp at each frame. According to the sampling timestamp, we set a 10Hz sampling timestamp and retrieve the multi-modal data frames that are closest to this timestamp. In this manner, we guarantee that the collected data are well-synchronized, with the synchronization error being within 25ms as the lowest sampling rate of all sensors is 20Hz (LiDAR).

## 4 Dataset

### 4.1 Subjects

Our human subject study is approved by the IRB at the Nanyang Technological University. The subject recruitment is voluntary, and the involved subject has been informed that the de-identified data was made publicly available for research purposes. The recruitment process is voluntary, and the experiments are conducted in several labs. Prior to participation, we provide detailed information to the subjects about the research goal, data collection procedure, potential risks, and the tutorial. The consent form is signed by every participating subject. Eventually, we recruit 40 human subjects locally in the university including 11 females and 29 males, with an average age of $25.3 \pm 2.8$, weight of $66.1 \pm 12.0$kg, height of $172.3 \pm 7.9$cm, and Body Mass Index (BMI) of $22.2 \pm 3.2$.

### 4.2 Categories of Human Motions

MM-Fi consists of 27 action categories which include 14 daily activities and 13 rehabilitation exercises. The daily activities are geared towards potential smart home and building applications, while the rehabilitation categories are designed to contribute to healthcare applications. Specifically, the daily activities include the common physical activities of various body parts: (a) chest expanding horizontally, (b) chest expanding vertically, (c) left side twist, (d) right side twist, (e) raising left arm, (f) raising right arm, (g) waving left arm, (h) waving right arm, (i) picking up things, (j) throwing toward left side, (k) throwing toward right side, (l) kicking toward left direction using right leg, (m) kicking toward right direction using left leg, (n) bowing. Meanwhile, the rehabilitation exercises are derived from [1] as (a) stretching and relaxing in free form, (b) mark time, (c) left upper limb extension, (d) right upper limb extension, (e) left front lunge, (f) right front lunge, (g) both upper limbs extension, (h) squat, (i) left side lunge, (j) right side lunge, (k) left limbs extension, (l) right limbs extension, (m) jumping up. These exercises are known to relieve pain, light up mood and reduce anxiety and fatigue. Each subject performs all 27 actions for a duration of 30 seconds.

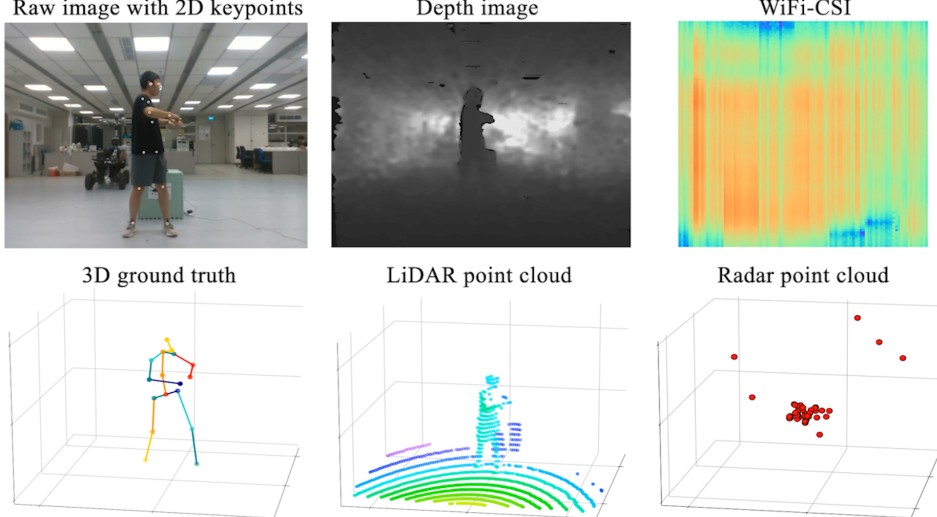

Figure 2: The visualization of five sensor modalities and 3D annotation in MM-Fi.

## 4.3 Data Annotation and Processing

**2D and 3D Human Pose Annotation.** We start by utilizing a large deep learning model, HRNet-w48 [39], to obtain the 2D keypoints $\mathcal{K}_{2D} = \{k_i \in \mathbb{R}^{17\times2}, i = 1, ..., N\}$ of $N$ frames from the two views, *e.g.*, two infra-red cameras. Then relying on the intrinsic and extrinsic parameters of cameras derived from the camera calibration, we triangulate 3D keypoints $\hat{\mathcal{P}}_{3D} = \{p_i \in \mathbb{R}^{17\times3}, i = 1, ..., N\}$ with the two views of 2D keypoints. However, due to the possibility of inaccurate triangulation, we further refine the 3D keypoints through an optimization process, which can be generally described as:

$$\min_{\hat{\mathcal{P}}_{3D}} \mathcal{L} = \mathcal{L}_G + \lambda_0 \mathcal{L}_A, \tag{1}$$

$$\mathcal{L}_G = \sum_{n=1}^{N} \left\{ \lambda_1 \sum_{c=1}^{C} \|f_c(p_n) - k_n^c\| + \lambda_2 \|p_{n+1} - p_n\| + \lambda_3 \sum_{j\in\Omega_\mathcal{B}} \|\mathcal{B}_{n,j} - \bar{\mathcal{B}}_j\| \right\}, \tag{2}$$

where $\mathcal{L}_G$ and $\mathcal{L}_A$ denote the general regularizer and action regularizer, respectively. In $\mathcal{L}_G$, the function $f_c(\cdot)$ projects the world coordinate to the pixel coordinate of the $c$th camera, parameterized by the corresponding intrinsic matrix, distortion coefficients and rotation-translation matrices. The first term of the loss function denotes the reprojection error of all $C$ cameras. The second term represents the smoothness loss that is introduced to reduce inter-frame outliers [1]. Moreover, for a specific subject, the bone (*e.g.*, leg) length $\{\mathcal{B}_j\}$ should remain constant regardless of the action, we thus apply a bone length constraint to the optimization process, where $\Omega_\mathcal{B}$ and $\bar{\mathcal{B}}_j$ denote the bone indices collection and the average length of $j$th bone for all frames [7] respectively. The triangulated 3D keypoints $\hat{\mathcal{P}}_{3D}$ serve as the initial value, and after optimization, we obtain $\hat{\mathcal{P}}_{3D} \to \mathcal{P}_{3D}$, which is considered as the ground truth for 3D keypoints. However, purely minimizing $\mathcal{L}_G$ does not optimize the estimated 3D joints well. For example, actions with turn-back and crouch may have occlusion, leading to incorrect recognition of some keypoints in the observed data regardless of the type of sensor utilized. Therefore, we set limitations on some joints based on the actions and bone lengths, which is why we introduce the action regularizer $\mathcal{L}_A$. The formation of $\mathcal{L}_A$ depends on the specific action and is described in the Appendix A.2.

**3D Position.** For mmWave radar and WiFi, an essential task is to estimate the precise 3D position of a subject with respect to the sensor, which serves many applications such as gaming and metaverse avatar simulation. To this end, we propose to additionally fuse the results of LiDAR data point and camera to provide the circumscribed cube of the subject. We annotate 2000 frames randomly sampled from MM-Fi to validate that the average error of the 3D position cube is within 50 mm.

**3D Dense Pose.** Recently, 3D human body modeling arouses more attention for AR and metaverse applications [20]. Dense pose estimation aims at mapping sensing data to the 3D surface of the human body [15], which is currently achieved by RGB. To enable wireless dense pose estimation based on radar, LiDAR, and WiFi, we provide the labels obtained from an advanced RGB-based dense estimation model [26] utilized in recent work for pose annotation [13].

**Temporal Action Segments.** Temporal action segmentation aims at densely identifying actions in long RGB sequences, and has become an increasingly essential task thanks to the growing numbers of long videos [12, 22, 42, 19]. Currently, temporal action segmentation is achieved through RGB. To enable temporal action segmentation based on modalities other than RGB (i.e., radar, LiDAR, and WiFi) and to provide more fine-grained samples, we provide the temporal segment labels obtained through a comprehensive segment annotation process. The details of the annotation process and annotation samples are presented in the Appendix.

**Post-Processing and Data Loader.** We unify the coordinate system for the point cloud data from LiDAR and mmWave radar according to the right-hand rule. To ensure the synchronized frame for all modalities, we construct the frame of MM-Fi dataset with a uniform sampling rate of 10Hz. Figure 2 shows one MM-Fi data frame. All the data except the temporal segments are saved using the *numpy* array format, *i.e.*, "npy" files, while the temporal segments are saved using a single ".csv" file. The PyTorch data loader is provided to load one or multiple modalities conveniently.

**Keypoints Quality** To evaluate the quality of our 3D keypoints, we first re-project them onto 2D keypoints and then manually annotate 100 frames for each action category, resulting in a total of 2,700 video frames. We compute the re-projection error as the percentage of correctly located keypoints using a threshold of 50% of the head segment length (denoted as PCKh@0.5). The analysis reveals that the re-projection PCKh@0.5 is 95.66%, indicating the high quality of our annotation.

**Intended Uses** MM-Fi opens up a wide range of applications in human sensing tasks, encompassing both existing and novel areas. By harnessing diverse modalities, it facilitates robust 2D/3D human pose estimation (HPE) across various modalities, paving the way for multi-sensor HPE through effective multi-modal learning techniques. Moreover, due to the inclusion of multiple subjects and environments, MM-Fi empowers new sensing and recognition tasks in wireless sensing. It enables self-supervised multi-modal sensing, revolutionizing the way data is leveraged across different modalities without relying on explicit annotations. MM-Fi empowers cross-domain HPE by developing new domain adaptation and domain generalization techniques, contributing to pose estimation across different domains and environments. Lastly, it enables few-shot action recognition by providing a rich dataset encompassing various actions and scenarios, empowering efficient and accurate recognition of actions with limited training samples. The versatility and richness of MM-Fi make it a valuable resource for researchers and practitioners in the field of human sensing.

## 5  Benchmark and Evaluation

In this section, we introduce the benchmark setup, evaluation metrics, and baseline methods for single-modal and multi-modal human pose estimation based on the proposed MM-Fi dataset. The results are analyzed to show the merits and drawbacks of these modalities. We also include a benchmark on skeleton-based action recognition [38, 9] using our dataset as a supplement in the Appendix.

### 5.1  Benchmark Setup

**Protocol.** We provide three protocols that are tailored to the various scenarios, based on the categories in Section 4.2. **Protocol 1 (P1)** includes 14 daily activities that are performed freely in space, *e.g.*, picking up things and raising arms. **Protocol 2 (P2)** includes 13 rehabilitation exercises that are performed in a fixed location, *e.g.*, limb extension. **Protocol 3 (P3)** includes all 27 activities. By using these three protocols, we evaluate the performance of the models on both free and fixed actions.

**Data Splits.** To evaluate the model, we provide three data split strategies. **Setting 1 (S1 Random Split)** involves a random split of all video samples into training and testing sets, with a split ratio of 3:1. **Setting 2 (S2 Cross-Subject Split)** splits the data by subject with 32 subjects for training and 8 subjects for testing. **Setting 3 (S3 Cross-Environment Split)** randomly selects 3 environments for training and 1 environment for testing.

**Evaluation Metrics.** We assess the performance of the models using two widely-used metrics in human pose estimation: Mean Per Joint Position Error (MPJPE) and Procrustes Analysis MPJPE (PA-MPJPE) [16]. MPJPE measures the difference between the ground truth and prediction for all joints by Euclidean distance after aligning the pelvis of the estimated and true 3D pose. PA-MPJPE refers to the MPJPE after adopting the Procrustes method [55] for alignment, which conducts a similarity transformation including translation, rotation, and scaling before the MPJPE calculation. MPJPE measures both the quality of joints and the spatial position, while PA-MPJPE focuses on the quality of joints. We provide the mean and standard deviation of these metrics across 3 runs.

**Baseline Methods.** We evaluate the model of 3D human pose estimation using a single modality including RGB image (I), LiDAR (L), mmWave radar (R), and WiFi (W) using the recent models. These methods are briefly introduced as follows:

- **RGB:** We use the RGB-based pose estimation model from [29] that transforms a sequence of 2D keypoints into 3D pose using a convolutional neural network. In the benchmark, we directly use its pre-trained model and evaluate it on our test sets.

- **LiDAR and mmWave:** For mmWave radar, we adopt the data processing technique from [2] that aggregates the consecutive frames into one data sample, while we directly use the vanilla LiDAR point cloud at each frame. Then we upgrade the neural network in [2] using the PointTransformer [51]. The model is trained from scratch using our training split.

- **WiFi:** We employ the data processing and model from MetaFi++ [56] with a convolutional network and some transformer blocks trained to regress WiFi CSI data to a root joint. The model is trained from scratch on our dataset.

We evaluate the performance of combining multiple modalities for various scenarios, including a robotics setting (I+L), low-cost setting (R+W), privacy-preserved setting (R+L+W), and all modality fusion (R+L+W+I). We adopt the simple result fusion method in the benchmark following the Least Mean Square (LMS) algorithm to learn the fusion weights of multiple sensors through linear combination [28] and leave space for future multi-modal fusion research. Instead of directly solving for the optimal weighting coefficients for each sensor, we reformulate the problem as a search for the optimal mean square estimation error, which allows for automatic parameter adjustment without the need for a correlation matrix.

### 5.2 Results and Analytics

**Random Split Results (S1).** Table 3 presents the evaluation results for human pose estimation using a single modality, including RGB, LiDAR, mmWave radar, and WiFi, under three settings and three protocols. Under random split (S1), LiDAR achieves 98.1, 94.9, and 92.5mm MPJPE for P1, P2, and P3, respectively, stably outperforming other modalities. The mmWave results are better than other modalities regarding the PA-MPJPE metric, achieving 55.6, 55.3, and 57.3mm PA-MPJPE for P1, P2, and P3, respectively. Due to the resolution limit of WiFi CSI, the results of WiFi modality are the worst. The RGB-based results are not satisfactory due to the domain gap, *i.e.*, the distribution shift between the dataset for the pre-trained model and our dataset.

**Cross-Subject Results (S2).** In S2, the focus of the evaluation is on testing the model's robustness against subject differences. Results in Table 3 show that the LiDAR and mmWave radar models show good generalization ability under S2. The PA-MPJPE results of LiDAR and mmWave radar models only vary within 3mm when they are compared to the S1 for three protocols. Nevertheless, the 3D HPE model via WiFi CSI shows a significant decline in performance thanks to the limited resolution which fails to fully capture human subtle motions and differences, thereby restricting the model's generalization ability across different subjects.

**Cross-Environment Results (S3).** In the cross-environment setting, the performances of all modalities significantly deteriorate. It is observed that 3D HPE based on mmWave radar achieves 166.2, 168.0, and 161.6mm MPJPE for P1, P2, and P3, respectively, significantly outperforming other modalities. A likely reason is that mmWave point cloud reflects the moving objects in space [32] and thus it is the least affected. The LiDAR MPJPE and PA-MPJPE drop a lot because the LiDAR data also captures many points on the ground and nearby objects. The WiFi CSI reflects the multi-path effect of the propagated signals, so the environment changes lead to decreasing performance for WiFi HPE model [58, 49].

|  |  | Protocol 1 | | Protocol 2 | | Protocol 3 | |
|---|---|---|---|---|---|---|---|
| Modality | Setting | MPJPE (mm) | PA-MPJPE (mm) | MPJPE (mm) | PA-MPJPE (mm) | MPJPE (mm) | PA-MPJPE (mm) |
| RGB | S1 | 263.3 | 80.0 | 291.6 | 83.6 | 279.0 | 81.2 |
|  | S2 | 267.7 | 81.2 | 304.3 | 82.6 | 285.3 | 81.9 |
|  | S3 | 276.6 | 83.0 | 301.5 | 85.3 | 288.6 | 84.1 |
| LiDAR | S1 | $98.1_{\pm2.2}$ | $65.2_{\pm0.7}$ | $94.9_{\pm1.1}$ | $60.4_{\pm1.8}$ | $92.5_{\pm0.6}$ | $61.5_{\pm1.3}$ |
|  | S2 | $110.1_{\pm2.9}$ | $66.2_{\pm1.2}$ | $103.9_{\pm0.6}$ | $60.3_{\pm2.3}$ | $103.8_{\pm1.5}$ | $61.5_{\pm1.2}$ |
|  | S3 | $192.3_{\pm30.4}$ | $100.4_{\pm5.4}$ | $186.0_{\pm2.9}$ | $103.5_{\pm11.9}$ | $303.8_{\pm11.6}$ | $133.0_{\pm3.3}$ |
| mmWave | S1 | $109.8_{\pm2.7}$ | $55.6_{\pm1.4}$ | $124.3_{\pm2.2}$ | $55.3_{\pm2.2}$ | $117.0_{\pm3.7}$ | $57.3_{\pm1.8}$ |
|  | S2 | $128.4_{\pm6.9}$ | $58.7_{\pm4.3}$ | $142.2_{\pm0.6}$ | $57.4_{\pm2.3}$ | $129.7_{\pm2.2}$ | $60.0_{\pm1.7}$ |
|  | S3 | $166.2_{\pm4.5}$ | $73.9_{\pm2.7}$ | $168.0_{\pm0.9}$ | $73.0_{\pm3.9}$ | $161.6_{\pm1.8}$ | $73.7_{\pm0.6}$ |
| WiFi | S1 | $186.9_{\pm0.1}$ | $120.7_{\pm0.9}$ | $213.5_{\pm0.5}$ | $121.4_{\pm0.1}$ | $197.1_{\pm0.6}$ | $121.2_{\pm0.5}$ |
|  | S2 | $222.3_{\pm0.8}$ | $125.4_{\pm2.3}$ | $247.0_{\pm0.2}$ | $122.7_{\pm1.3}$ | $231.1_{\pm0.4}$ | $124.0_{\pm0.5}$ |
|  | S3 | $367.8_{\pm0.9}$ | $121.0_{\pm2.2}$ | $360.2_{\pm1.3}$ | $117.2_{\pm0.9}$ | $369.5_{\pm0.3}$ | $116.0_{\pm1.8}$ |

Table 3: 3D human pose estimation results for RGB, LiDAR, mmWave radar, and WiFi. The mean and standard deviation of MPJPE are reported under 3 settings and 3 protocols.

|  |  | Protocol 1 | | Protocol 2 | | Protocol 3 | |
|---|---|---|---|---|---|---|---|
| Modalities | Setting | MPJPE (mm) | PA-MPJPE (mm) | MPJPE (mm) | PA-MPJPE (mm) | MPJPE (mm) | PA-MPJPE (mm) |
| I+L | S1 | $94.3_{\pm1.5}$ | $63.9_{\pm0.6}$ | $90.1_{\pm1.8}$ | $58.9_{\pm2.1}$ | $87.5_{\pm0.8}$ | $59.8_{\pm1.2}$ |
|  | S2 | $109.4_{\pm1.6}$ | $65.8_{\pm0.6}$ | $103.5_{\pm0.6}$ | $60.2_{\pm2.4}$ | $104.6_{\pm0.6}$ | $62.1_{\pm0.3}$ |
|  | S3 | $159.6_{\pm25.9}$ | $84.5_{\pm1.8}$ | $153.5_{\pm3.37}$ | $87.1_{\pm10.3}$ | $192.1_{\pm21.8}$ | $89.6_{\pm3.9}$ |
| R+W | S1 | $94.2_{\pm3.7}$ | $55.2_{\pm6.3}$ | $92.8_{\pm13.2}$ | $51.8_{\pm4.2}$ | $113.2_{\pm4.2}$ | $75.6_{\pm2.8}$ |
|  | S2 | $127.4_{\pm2.8}$ | $72.9_{\pm4.0}$ | $151.5_{\pm4.4}$ | $64.6_{\pm1.9}$ | $127.5_{\pm5.8}$ | $77.3_{\pm5.1}$ |
|  | S3 | $104.9_{\pm9.3}$ | $47.6_{\pm2.4}$ | $110.5_{\pm10.1}$ | $54.2_{\pm1.7}$ | $116.1_{\pm7.8}$ | $57.2_{\pm4.9}$ |
| R+L+W | S1 | $74.1_{\pm1.7}$ | $46.7_{\pm1.4}$ | $72.7_{\pm6.8}$ | $42.7_{\pm1.1}$ | $72.7_{\pm6.8}$ | $42.7_{\pm1.1}$ |
|  | S2 | $111.9_{\pm3.6}$ | $83.0_{\pm3.6}$ | $106.5_{\pm1.1}$ | $63.5_{\pm2.2}$ | $99.6_{\pm1.5}$ | $72.0_{\pm1.5}$ |
|  | S3 | $119.0_{\pm3.8}$ | $66.8_{\pm5.8}$ | $111.1_{\pm7.8}$ | $61.8_{\pm6.8}$ | $126.3_{\pm17.9}$ | $61.0_{\pm10.8}$ |
| R+L+W+I | S1 | $72.9_{\pm1.0}$ | $47.7_{\pm1.2}$ | $69.5_{\pm3.4}$ | $43.1_{\pm1.9}$ | $89.8_{\pm2.0}$ | $63.2_{\pm1.9}$ |
|  | S2 | $112.0_{\pm4.7}$ | $82.8_{\pm4.0}$ | $114.8_{\pm2.32}$ | $67.0_{\pm2.73}$ | $99.2_{\pm1.0}$ | $73.0_{\pm1.4}$ |
|  | S3 | $98.1_{\pm10.4}$ | $58.3_{\pm8.8}$ | $92.7_{\pm11.65}$ | $65.0_{\pm7.59}$ | $116.5_{\pm23.6}$ | $70.6_{\pm13.5}$ |

Table 4: 3D human pose estimation results of multi-sensor fusion, where the modalities are denoted as I (Image), L (Lidar), R (mmWave Radar) and W (WiFi CSI).

**Modality Fusion.** We search for the optimal weights and obtain the improved multi-modal HPE results in Table 4 compared to the single-modal results in Table 3. For example, the fusion of RGB images and LiDAR (I+L) outperforms LiDAR significantly for MPJPE under three settings. For all the activities (P3), the best PA-MPJPE results come from the modality fusion of R+L+W, achieving 42.7, 72.0, and 61.0mm for S1, S2, and S3 respectively. This demonstrates that in real-world scenarios, fusing multiple modalities in some ways for 3D HPE can be more robust and accurate.

# 6 Limitations of MM-Fi

The MM-Fi dataset currently has limitations regarding annotations and benchmarks. Firstly, the annotation process is manual and the quality is limited. Due to the resolution of wireless sensing data, the current sensing tasks are restricted to activity level and keypoint level, which have been validated with high quality. However, for tasks that require higher resolution, such as dense pose estimation, we only provide annotations obtained from algorithms that have not been validated. These annotations are included to facilitate new tasks and inspire further research. In the upcoming MM-Fi V2.0, we plan to use a motion capture system to annotate the dense pose to address this limitation. Secondly, as the first dataset to offer mmWave radar, LiDAR, RGBD, and WiFi data simultaneously, there are tasks that have not been extensively studied yet. Therefore, some baseline methods developed by us may not perform optimally without careful design. Our intention is to inspire researchers to explore these unexplored fields and contribute to the future benchmarking of the MM-Fi dataset. Thirdly, the current dataset is collected in a controlled condition, i.e., 3m away and same facing direction, with a single person. We plan to include multi-orientation, multi-location, and multi-user scenarios in MM-Fi 2.0. We acknowledge these limitations and aim to improve the dataset by addressing them in future versions. Our objective is to provide a comprehensive resource that encourages research and advancements in the non-intrusive wireless sensing research community.

# 7 Conclusion

In this paper, we present *MM-Fi*, a novel non-intrusive multi-modal 4D human pose dataset for wireless human sensing. It consists of four sensing modalities: RGB-D, LiDAR, mmWave radar, and WiFi, and is currently the most comprehensive benchmark for wireless human pose estimation. Our MM-Fi is composed of 320.76k synchronized frames, and 27 categories of poses performed by 40 subjects. The collection process and the four data modalities are introduced, followed by extensive experiments to produce a single-modal and multi-modal benchmark. Our proposed dataset contributes to the multi-modal human pose estimation research and helps researchers choose suitable sensor modalities according to their advantages and disadvantages. We hope that our proposed MM-Fi dataset and benchmarks can facilitate future research in machine learning, ubiquitous computing, computer vision, mobile computing, and healthcare.

**Acknowledge.** This work is supported by NTU Presidential Postdoctoral Fellowship, "Adaptive Multimodal Learning for Robust Sensing and Recognition in Smart Cities" project fund, at Nanyang Technological University, Singapore.

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
