# MM-Fi: Multi-Modal Non-Intrusive 4D Human Dataset for Versatile Wireless Sensing

**Jianfei Yang**[1]*, **He Huang**[1], **Yunjiao Zhou**[1], **Xinyan Chen**[1], **Yuecong Xu**[1],
**Shenghai Yuan**[1], **Han Zou**[1], **Chris Xiaoxuan Lu**[2], **Lihua Xie**[1]
[1]School of Electrical and Electronic Engineering, Nanyang Technological University
[2]School of Informatics, University of Edinburgh
Project Page: https://ntu-aiot-lab.github.io/mm-fi
Dataset Toolbox: https://github.com/ybhbingo/MMFi_dataset

## A    Supplementary Material

### A.1    Ethics Statement

The MM-Fi human subject study in this paper has been reviewed and approved by the IRB committee at the Nanyang Technological University (IRB-2022-1067). The MM-Fi data has been de-identified by facial blur. The subject recruitment is voluntary, and the involved subject has been informed that the de-identified data was made publicly available for research purposes. As far as we know, this research does not endanger any person directly. Nevertheless, it is acknowledged that pose estimation and activity recognition research can potentially be used with malicious intent, such as user behavior monitoring.

### A.2    Dataset Documentation

**Modality and Action Category.**    The MM-Fi dataset contains six modalities including RGB, infra, depth, mmWave, LiDAR and WiFi CSI with open and widely used data formats, and consists of 4 environments for domain diversity. Demonstration videos can be referred to our project page (https://ntu-aiot-lab.github.io/mm-fi). 40 volunteers participated in the data collection progress with 10 volunteers in one environment. The recommended actions contain daily actions and rehabilitation actions, which are summarized in Table 1. Besides, illustration of all actions by a volunteer can be viewed in the action checklist (*i.e.*, demo image sequence of each action).

**Distribution and Maintenance.**    The dataset with recommended instructions on how to download and use is maintained in our GitHub repository (https://github.com/ybhbingo/MMFi_dataset) with a registered DOI, which we will update timely according to the users' and community's advice. In addition, for the accessibility of dataset and long-term preservation, we have uploaded dataset to Google Drive, Baidu Netdisk and Alibaba Cloud with long-term cloud storage service (links are shared in the GitHub repository). We provide two ways of downloading the dataset: whole zip file download and multi-split download. The original dataset is organized in a clear structure so that users could check the recovered dataset conveniently, which can be shown by Figure 2.

**Dataset Toolbox.**    For convenient data loading, we have transformed the sensing data from different modalities into the open and widely used data formats, which are listed in Table 2. We also developed the dataset toolbox in the GitHub repository (https://github.com/ybhbingo/MMFi_dataset) that provides the dataloader for PyTorch deep learning framework. The users can download the data from the link and follow the instructions in our GitHub repository to load the data easily. The

---

*Corresponding authors (yang0478@e.ntu.edu.sg)

| Action | Description | Category | Action | Description | Category |
|--------|-------------|----------|--------|-------------|----------|
| A01 | Stretching and relaxing | Rehabilitation activities | A15 | Lunge (toward left) | Rehabilitation activities |
| A02 | Chest expansion (horizontal) | Daily activities | A16 | Lunge (toward right) | Rehabilitation activities |
| A03 | Chest expansion (vertical) | Daily activities | A17 | Waving hand (left) | Daily activities |
| A04 | Twist (left) | Daily activities | A18 | Waving hand (right) | Daily activities |
| A05 | Twist (right) | Daily activities | A19 | Picking up things | Daily activities |
| A06 | Mark time | Rehabilitation activities | A20 | Throwing (toward left) | Daily activities |
| A07 | Limb extension (left) | Rehabilitation activities | A21 | Throwing (toward right) | Daily activities |
| A08 | Limb extension (right) | Rehabilitation activities | A22 | Kicking (toward left) | Daily activities |
| A09 | Lunge (toward left-front) | Rehabilitation activities | A23 | Kicking (toward right) | Daily activities |
| A10 | Lunge (toward right-front) | Rehabilitation activities | A24 | Body extension (left) | Rehabilitation activities |
| A11 | Limb extension (both) | Rehabilitation activities | A25 | Body extension (right) | Rehabilitation activities |
| A12 | Squat | Rehabilitation activities | A26 | Jumping up | Rehabilitation activities |
| A13 | Raising hand (left) | Daily activities | A27 | Bowing | Daily activities |
| A14 | Raising hand (right) | Daily activities | | | |

Table 1: The action list including daily activities and rehabilitation activities.

| Modality | Data format | Data size | File extension | Modality | Data format | Data size | File extension |
|----------|-------------|-----------|----------------|----------|-------------|-----------|----------------|
| RGB | Numpy array | $297 \times 17 \times 2$ | .npy | WiFi CSI | MATLAB matrix | $297 \times 3 \times 114$ | .mat |
| infra1 | Numpy array | $297 \times 17 \times 2$ | .npy | infra2 | Numpy array | $297 \times 17 \times 2$ | .npy |
| mmWave | Binary | 297 clouds | .bin | LiDAR | Binary | 297 clouds | .bin |
| Depth | Image | $297 \times 640 \times 480$ | .png | | | | |

Table 2: The data formats for different modalities.

anonymized version of RGB images, preserving the identifiable information of volunteers (already with permissions from all volunteers), can be accessed after the application forms.

**Environmental Details.** To enhance the diversity of the MM-Fi dataset, we acquired data from four distinct environmental settings (referred to as E01, E02, E03, and E04). Within each environment, a cohort of ten participants was enlisted to execute a series of 27 commonplace human activities. As shown in Fig. 1, the sensor platform is deployed both horizontally and vertically within Room 1 and Room 2, respectively, contributing to 4 environmental settings. While both rooms share identical dimensions (8.5 meters in length and 7.8 meters in width), they exhibit distinct spatial layouts, which leads to diverse and varied sensor signals.

**Data Processing.** It is worth noting that for modalities with fixed input size from sensors, we have reformed the sensing data into the common structured formats, i.e. an array structure in Numpy or MATLAB, as shown in Table 2. For LiDAR data, since the original point cloud contains redundant information (the laser scanner provides reflected data for all directions in our LiDAR device, Liu but we only focus on the areas of interests where the subject stands), we have filtered the whole cloud using bounding boxes. While for the mmWave modality, the number of points varies with the body movement, indicating that even two consecutive frames would likely have different data sizes. To enhance the sensing quality, we have aggregated five adjacent frames into a new frame for use. Furthermore, padding is adopted for mmWave and LiDAR point clouds produced by the PyTorch dataloader, in which the padded size is determined by the largest sample size within the batch. For WiFi CSI data, there are some "-inf" values in some sequences. The "-inf" number comes from the noise or empty frames from the CSI tool. In our benchmark, we deal with these numbers by linear interpolation. To facilitate the users, we have embedded these processing codes into our dataset tool. When the user loads our WiFi CSI data, these numbers will be handled by linear interpolation. The codes can be found here.

**Temporal Segment Annotation Process.** As presented in Section 4.3, we provide the temporal segment labels to enable temporal action segmentation and to provide more fine-grained samples. The segment annotation process is performed by human annotators with an automated segment annotation

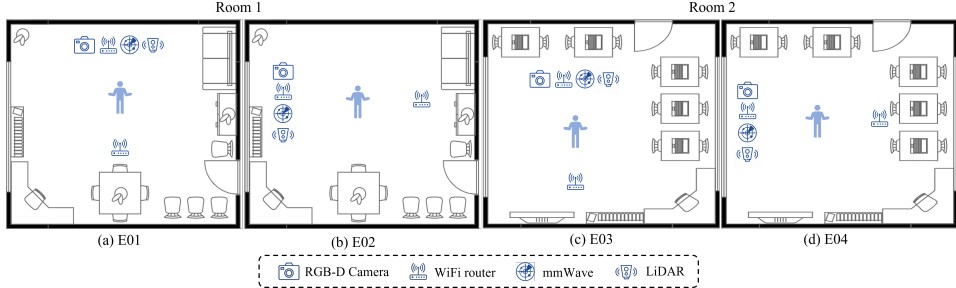

Figure 1: The illustration of the four environmental settings. The experiment is carried out in two rooms (Room 1 and Room 2), whose length and width are both 8.5m and 7.8m respectively.

program that feeds the human annotators with the long sequence frame-by-frame, and enables the generation of temporal segments through human selected breakpoints. Each breakpoint is essentially a frame that implies the end of an action. Each sequence is annotated by at least 5 human annotators. To ensure correct segment annotation, a voting process is performed after all human annotations are collected, where the breakpoints that are selected by the majority of the annotators are viewed as the ground-truth breakpoints. The resulting action segments are frames between each breakpoint. The resulting fine-grained samples are recorded in a ".csv" file with sample records as shown in Table 3. The records includes the Environment, Subject, Action, and Segments information, where the different segments are represented by the beginning and end frames.

| Environment | Subject | Action | Segments |
|---|---|---|---|
| E01 | S01 | A02 | 1-24;  25-50;  51-78;  79-105;  106-132;  133-160;  161-188;  189-218;  219-247;  248-277;  278-297 |
| E01 | S03 | A18 | 1-15; 16-28; 29-41; 42-54; 55-67; 68-82; 83-96; 94-110; 111-124; 125-139; 140-152; 153-167; 168-181; 182-196; 197-210; 211-225; 226-239; 240-254; 255-267; 268-282; 283-297 |

Table 3: The data formats for different modalities.

**Additional Annotator Details.** Besides the regular loss function for annotating optimization, there usually exists body occlusion in a series of actions, which would cause failure of keypoints recognition and thus lead to inaccurate ground truth. As a result, we have introduced the specific regularizer $\mathcal{L}_A$ for better annotation quality with expert knowledge from several action directors and volunteers. To be specific, we take the action "throwing toward left" (A20) for instance to illustrate how we define $\mathcal{L}_A$ and handle the occlusion problem. When a volunteer faces toward the left side, the left part of the body (including the left arm and the left hip joint) would not be observed by the sensors, so we put more regularizer terms regarding the relative positions between the joints, which is denoted by:

$$\mathcal{L}_A = \sum_{n=1}^{N} \left\{ \gamma_1 \left\| d(p_{n,lp}, p_{n,rp}) - l_p \right\| + \gamma_2 \left\| d(p_{n,ls}, p_{n,rs}) - l_s \right\| + \gamma_3 h(p_n) \right\}, \tag{1}$$

where $p_{n,lp}$ and $p_{n,rp}$ denote the left and right hip joint of $n$th frame, respectively. $l_p$ is the mean hip distance for the specific volunteer. The first term in (1) regularizes the unobservable left hip. Similarly, the second term is designed for the unobservable shoulder joint. The last term plays the role of constraining all the joints so that they would be within the sensing area, and the arms and legs would not be "bent" backward given the particular coordinates.

### A.3   License Agreement

This work has been licensed under CC BY-NC 4.0, and we bear all responsibility in case of violation of rights, the novelty of work, and privacy leaks.

### A.4   Benchmark Implementation Details

We implemented all the baseline methods using the PyTorch framework and ensured that the hyperparameter settings matched those specified in the original papers. All the codes with a manual (README) for benchmarking each modality can be checked in the link. To account for variability, we conducted each experiment three times with different random seeds and reported the mean and standard deviation of the results. Our experiments were performed on a local Ubuntu 20.04 server

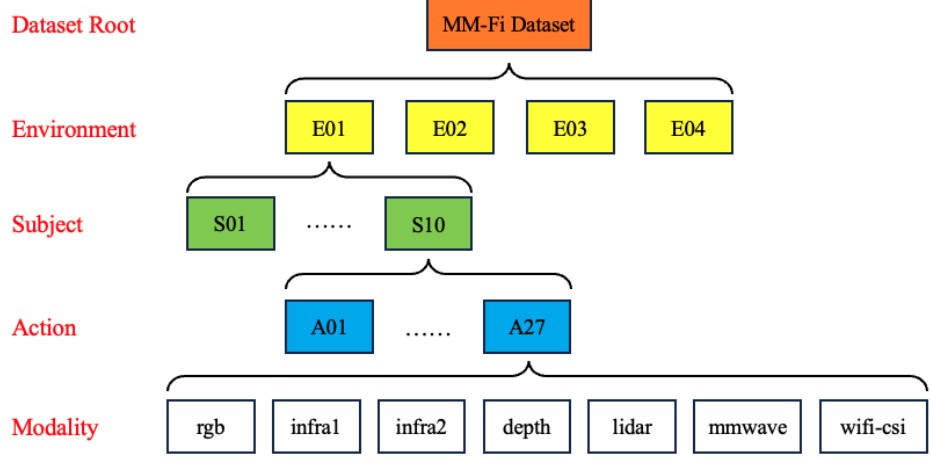

Figure 2: The expanded directory of original MM-Fi dataset.

equipped with 4 NVIDIA RTX 3090 GPUs, an AMD Ryzen PRO 3000WX Series Processor (64 Cores), and 128GB RAM. The learned weights for multi-modal fusion are summarized in Table 4.

### A.5 Details of Baseline Models.

**WiFi** We utilize MetaFi++ as our baseline, which is based on a convolutional network and employs a self-attention mechanism (in transformer) to learn the importance between different antennas. This enables selective fusion of relevant information from antennas, resulting in enhanced spatial capture capability and achieving state-of-the-art scores. There are approaches in the field of WiFi pose estimation, e.g., GoPose, WiPose, and Winect, all of which only use convolutional methods. Moreover, these methods handle WiFi data with only 30 subcarriers, while our MM-Fi has a higher granularity of data with 114 subcarriers, so the inconsistency of data prevents us from using these methods as baselines.

**LiDAR and mmWave radar** The current human pose estimation (HPE) methods for point cloud consist of convolutional networks, graph neural network, and the recent transformer. We chose Point Transformer [1] as our point cloud backbone because it shows state-of-the-art performance on many point cloud recognition tasks. Its design utilizes the self-attention mechanism to minimize the computational complexity for each layer and maximize the number of parallelizable calculations, which perfectly matches the characteristic of point cloud data.

**RGB** The visual baseline (VideoPose3D) is a popular 2D-to-3D solution that transforms 2D keypoints to the 3D keypoints, developed by Facebook Research. We choose this baseline since it has been widely utilized in many human pose estimation benchmarks [1][2]. There are many other works in computer vision for 3D human pose estimation, and we welcome the community to supplement more benchmarking results on our dataset.

### A.6 Additional Evaluation of RGB Model

Due the the difference of viewing angles between the Human3.6M [16] and our dataset, the pre-trained model on Human3.6M does not achieve satisfactory performance on the MPJPE metric, though the PA-MPJPE is significantly reduced due to the coordinates alignment. For better evaluation of the MM-Fi's RGB modality, the VideoPose3D [29] model has been further trained from scratch based on the MM-Fi's RGB modality. To be detailed, we use 81 2D-keypoint frames to generate one 3D-joint frame on the equivalent of a padding size of 40. The Adam optimizer is adopted and the learning rate decays from 0.001 gradually along epochs. The results are summarized in Table 5. We can observe that the re-trained RGB model obtains the best performance compared to other modalities.

| Modalities | Setting | Protocol 1 weights | Protocol 2 weights | Protocol 3 weights |
|---|---|---|---|---|
| I+L | S1 | [0.0432, 0.9568] | [0.0553, 0.9447] | [0.0614, 0.9386] |
| | S2 | [0.0131, 0.9869] | [-0.0013, 1.0013] | [-0.0230, 1.0230] |
| | S3 | [0.2634, 0.7366] | [0.2727, 0.7273] | [0.4957, 0.5043] |
| R+W | S1 | [0.7955, 0.2045] | [1.0126, -0.0126] | [0.8197, 0.1803] |
| | S2 | [0.9617, 0.0383] | [0.8124, 0.1876] | [0.9287, 0.0713] |
| | S3 | [0.8786, 0.1214] | [0.9459, 0.0541] | [0.9166, 0.0834] |
| R+L+W | S1 | [0.2146, 0.8506, -0.0652] | [0.2251, 0.7902, -0.0053] | [0.2251, 0.7802, -0.0053] |
| | S2 | [0.1494, 0.9645, -0.1139] | [0.2251, 0.7802, -0.0053] | [0.0664, 0.9434, -0.0098] |
| | S3 | [0.4307, 0.7733, -0.2040] | [0.7483, 0.3627, -0.1110] | [0.8843, 0.1852, -0.0695] |
| R+L+W+I | S1 | [0.1391, 0.9152, -0.0913, 0.0370] | [0.1569, 0.8247, -0.0240, 0.0424] | [0.0822, 0.9123, -0.0550, 0.0605] |
| | S2 | [0.1266, 0.9721, -0.1256, 0.0269] | [0.0529, 0.9523, 0.0618, -0.0670] | [0.0877, 0.9384, -0.0363, 0.0102] |
| | S3 | [0.2319, 0.5135, 0.0378, 0.2167] | [0.3965, 0.4997, -0.0761, 0.1799] | [0.4144, 0.3995, -0.1052, 0.2913] |

Table 4: The weight matrix for the multi-modal human pose estimation results. The weight order in the weights column corresponds to that in the modalities column.

| | Protocol 1 | | Protocol 2 | | Protocol 3 | |
|---|---|---|---|---|---|---|
| Setting | MPJPE (mm) | PA-MPJPE (mm) | MPJPE (mm) | PA-MPJPE (mm) | MPJPE (mm) | PA-MPJPE (mm) |
| S1 | $64.7_{\pm0.5}$ | $34.8_{\pm0.1}$ | $68.5_{\pm1.4}$ | $31.5_{\pm0.1}$ | $60.5_{\pm0.4}$ | $32.5_{\pm0.2}$ |
| S2 | $88.4_{\pm1.1}$ | $35.8_{\pm0.1}$ | $82.8_{\pm1.5}$ | $32.1_{\pm0.1}$ | $85.7_{\pm0.5}$ | $33.4_{\pm0.1}$ |
| S3 | $104.8_{\pm0.7}$ | $40.4_{\pm0.1}$ | $82.8_{\pm1.5}$ | $32.1_{\pm0.1}$ | $85.7_{\pm0.5}$ | $33.4_{\pm0.1}$ |

Table 5: The performance of the re-trained RGB model on the MM-Fi dataset. The mean and standard deviation of MPJPE are reported under 3 settings and 3 protocols.

## A.7 Visualization

To intuitively show the HPE results of each modality, we visualize the results of HPE using four modalities in Figure 3. It is shown that the results of RGB, LiDAR, and mmWave radar are quite accurate, but WiFi-based HPE is not satisfactory due to the resolution limitation.

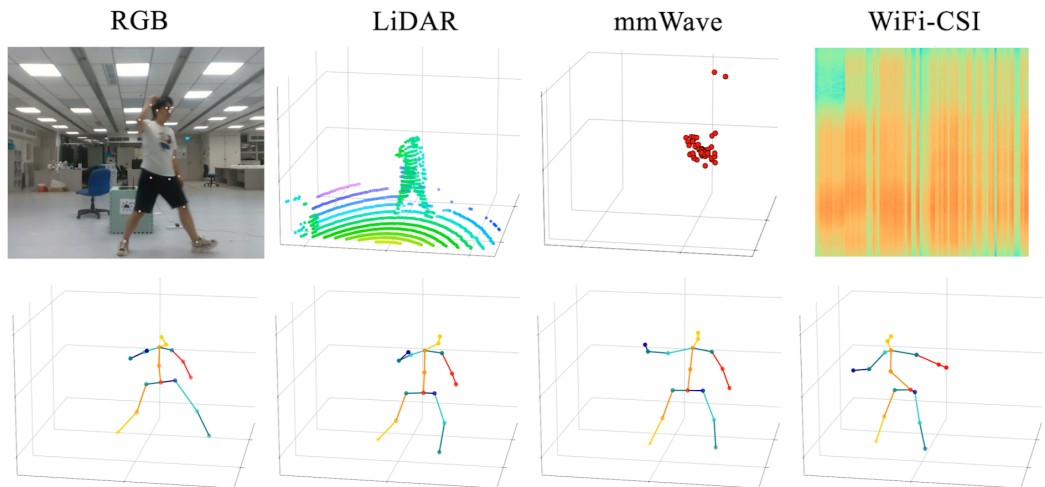

Figure 3: The visualization of human pose estimation using four modalities.

## A.8 Benchmark on Skeleton-Based Action Recognition

The MM-Fi can also serve the skeleton-based action recognition task based on various data modalities. Here we leverage the results obtained by human pose estimation to conduct a new benchmark.

**Implementation Details.** In the official split of HPE, we have 30 subjects for training and 10 subjects for testing. After HPE, we have 3D human joints of 10 subjects, which serve as the dataset

for skeleton-based action recognition using various data modalities. Each sequence is divided by 30 frames to generate small clips for action recognition. We split the data into 7 subjects (11340 frames) for training and 3 subjects (4860 frames) for testing. We leverage two novel skeleton-based action recognition methods, AGCN [2] (CVPR'19) and CTRGCN [1] (CVPR'21), with their pre-trained model parameters on NTU-RGBD datasets. The hyper-parameters for training follow the original papers. The learning rate starts from 0.1 and decays by 10 at the 5th and 10th epochs. The models are optimized by the SGD optimizer with a Nesterov momentum of 0.9. The models are trained for 20 epochs with a batch size of 32 and a weight decay of 0.0001.

**Results.**    As shown in Table 6, we can see that both AGCN and CTRGCN perform the best using RGB data since the two models are originally built upon RGB data and the human joints predicted by RGBD are quite well in the manuscript (Table 3). The second best modality is the mmWave radar point cloud, which is better than the LiDAR results. However, in Table 3 of the manuscript, it is shown that the HPE results of LiDAR are better than those of mmWave. We think that the better action recognition results of mmWave are caused by a more smooth prediction of sequences. The WiFi data cannot enable action recognition, as we find that some frames of joints predicted by WiFi are not robustly consistent. Another interesting finding is that CTRGCN performs better than AGCN on public RGBD datasets, but for various data modalities in MM-Fi, AGCN performs better. This implies that CTRGCN overfits some RGBD datasets and may not really generalize well on other datasets.

|  | AGCN [2] | | CTRGCN [1] | |
| --- | --- | --- | --- | --- |
|  | Top 1 (%) | Top 5 (%) | Top 1 (%) | Top 5 (%) |
| WiFi | 5.45 | 21.63 | 5.12 | 23.33 |
| mmWave Radar | 65.25 | 93.44 | 60.97 | 91.07 |
| LiDAR | 54.44 | 91.03 | 35.97 | 74.12 |
| RGBD | 87.78 | 99.42 | 66.98 | 93.60 |

Table 6: The benchmark results for skeleton-based action recognition.

## A.9    More Uses: New Research Tasks

MM-Fi significantly expands the horizons of research by furnishing meticulously synchronized multimodal human sensing data. As advised by the reviewer, we have encapsulated these novel research directions as follows, and have incorporated these enhancements into the manuscript:

**Cross-domain wireless sensing.**    The realm of wireless sensing is often challenged by recognition performance disparities stemming from domain shifts, which arise due to variations in environments and subjects. While this issue has garnered extensive exploration within action recognition based on RGB data, it remains relatively uncharted within the context of Human Pose Estimation (HPE) utilizing modalities such as mmWave, LiDAR, and WiFi. The MM-Fi dataset presents a unique opportunity to address this gap by facilitating research into domain adaptation and generalization for HPE on these emerging modalities. By enabling investigations into the adaptation of models across diverse domains, MM-Fi opens avenues for enhancing the robustness and applicability of wireless sensing technologies.

**Cross-modal supervision for fine-grained wireless sensing.**    Previous works have showcased the utility of WiFi and mmWave radar for action recognition tasks. However, MM-Fi introduces a transformative dimension by offering meticulously synchronized multimodal data and comprehensive annotations. Based on MM-Fi, cross-modal learning can enable these sparse data modality to achieve fine-grained recognition tasks, e.g., human pose estimation and action segmentation.

**Multi-modal wireless sensing.**    MM-Fi's integration of five distinct modalities empowers researchers to explore the potential of multi-modal wireless sensing, where different sensing technologies complement each other to achieve more comprehensive and accurate insights into human behavior. For example, integrating RGB and WiFi can overcome the illumination issue of RGB-based solutions. This dataset serves as a launchpad for pioneering investigations into techniques that fuse information from LiDAR, mmWave radar, and WiFi signals to attain a holistic understanding of human actions and interactions. As researchers delve into multi-modal fusion methods, the MM-Fi

dataset becomes a valuable resource for the development of advanced solutions that leverage the strengths of each modality, while compensating for their individual limitations.

### A.10  Sensor Specifications

We briefly introduce the sensors used in our collection platform. Note that some sensors have very high sampling rate or data resolution, but we do not use the maximum setting in our data collection. The experimental setting has been illustrated in Section 3.

**WiFi**   We develop a customized OpenWrt firmware for COTS WiFi devices (TP-Link N750) using the Atheros CSI tool to enable a large-scale implementation of various CSI-enabled applications. Our platform reports all the 114 subcarriers for the 40 MHz bandwidth on each antenna pair operating on 5 GHz. The platform has 3 pairs of antennas with one on transmitter and three on receivers to collect the CSI data based on our developed firmware tool, which finally provides a CSI data stream of 100Hz after average sliding on the raw data.

**Lidar**   Ouster OS1 32-channel LiDAR is used to acquire dense point cloud data. It contains 32 vertical beams, which provides $\pm$0.7-5cm vertical angular resolution. Its vertical field of view is 45 degrees and its range is 120m. It can capture dense point clouds with a maximum of 1,310,720 points per second. In MM-Fi, we collected the raw data under 10 Hz frequency.

**mmWave radar**   The Texas Instruments (TI) IWR6843AOP mmWave radar is used to collect mmWave point clouds with up to 30FPS. It is an integrated single-chip mmWave sensor that has 3 antennas to transmit FMCW and use 4 antennas to receive reflected FMCW by utilizing 60-64 GHz radio band. Under the condition of 1.0-V internal LDO bypass model and 48% duty cycle, its typical power consumption is 1.75 W.

**RGB-D**   The Intel RealSense camera D435, consisting of a depth module, an RGB module and a Realsense vision processor D4, is a stereo solution, offering quality depth for a variety of applications. The RGB module applies the rolling shutter technology, enabling the maximal 1920x1080 high-resolution RGB frame acquisition at a sensing rate of 30 fps. Its horizontal and vertical fields of view are 69 degrees and 42 degrees, respectively. The depth module, formed with one IR projector and two imagers, could obtain a larger sensing field of view with 87 (horizontal) and 58 (vertical) degrees, and provide a 1280x720 resolution depth frame at up to 90 fps. What's more, due to the global shutter technology, the depth module could even work in low-light situations, making the Realsense D435 a good solution for all-day depth sensing applications.