# OpenReview forum: "MM-Fi: Multi-Modal Non-Intrusive 4D Human Dataset for Versatile Wireless Sensing"
_NeurIPS.cc/2023/Track/Datasets_and_Benchmarks — NeurIPS 2023 Datasets and Benchmarks Poster_

### Official Review · Reviewer_2naR · 2023-07-17
**The paper proposes a new multi-modal non-intrusive 4D human dataset to facilitate the algorithm development of wireless human sensing. The topic is relevant and the dataset holds potential value, critical issues need to be addressed.**

**Rating:** 7
**Confidence:** 5
**Correctness:** 1.	In line 186, L_G and L_A should de…
**Clarity:** The paper is well written. The contri…

**Strengths:**

1.	The dataset proposed in this paper is the first human dataset to integrate RGB-D, LiDAR, mmWave radar, and WiFi modalities.
2.	LiDAR, mmWave radar, and WiFi data in the dataset enable more privacy-preserving human sensing applications.


**Additional Feedback:**

Can the author show a video to demonstrate the quality of their dataset?

**Documentation:**

There is sufficient detail on data collection and organization, availability and maintenance, and ethical and responsible use. For benchmarks, there is sufficient detail to support reproducibility.

**Limitations:**

1.	It is unclear about the quality of the dataset. This work uses 2D positions estimated by HRNet, and uses 2D positions estimated from two views to generate 3D positions. However, the correctness of the annotation process remains ambiguous.
2.	The motivation of the work is unclear. It is similar to mRI[1], which published in NeurIPS 2022 dataset track. The major differences are the introduction of the WIFI modality. However, due to the resolution limitations of wireless sensor data, the performance of HPE is dropped according to the experimental results. It is unclear why we need WIFI signal alongside RGB,LiDAR and mmWave.
3.	The design and results of multi-modal experiments do not fully reflect the advantages of WIFI.
4.	Several related HPE datasets are missing from the discussion, including mmbody[2], LIPD[3], LiCamPose[4], SLOPER4D[5], CIMI4D[6].

**References **

[1] Sizhe An, Yin Li, and Umit Ogras. mRI: Multi-modal 3d human pose estimation dataset using mmwave, RGB-d, and inertial sensors. In Thirty-sixth Conference on Neural Information Processing Systems Datasets and Benchmarks Track, 2022.
[2] Chen, Xiangyu Wang, Shaohao Zhu, Yanxu Li, Jiming Chen, and Qi Ye. mmbody benchmark: 3d body reconstruction dataset and analysis for millimeter wave radar. In Proceedings of the 30th ACM International Conference on Multimedia, pages 3501–3510, 2022.
[3] Chengfeng Zhao, Yiming Ren, Yannan He, Peishan Cong, Han Liang, Jingyi Yu, Lan Xu, and Yuexin Ma. Lidar-aid inertial poser: Large-scale human motion capture by sparse inertial and lidar sensors. arXiv preprint arXiv:2205.15410, 2022.
[4] Peishan Cong, Yiteng Xu, Yiming Ren, Juze Zhang, Lan Xu, Jingya Wang, Jingyi Yu, and Yuexin Ma. Weakly supervised 3d multi-person pose estimation for large-scale scenes based on monocular camera and single lidar. arXiv preprint arXiv:2211.16951, 2022.
[5] Yudi Dai, Yitai Lin, XiPing Lin, Chenglu Wen, Lan Xu, Hongwei Yi, Siqi Shen, Yuexin Ma, 314 and Cheng Wang. Sloper4d: A scene-aware dataset for global 4d human pose estimation in 315 urban environments. In Proceedings of the IEEE/CVF Conference on Computer Vision and 316 Pattern Recognition, pages 682–692, 2023.
[6] Ming Yan, Xin Wang, Yudi Dai, Siqi Shen, Chenglu Wen, Lan Xu, Yuexin Ma, and Cheng Wang. Cimi4d: A large multimodal climbing motion dataset under human-scene interactions. In Proceedings of the IEEE/CVF Conference on Computer Vision and Pattern Recognition, 463 pages 12977–12988, 2023.


**Opportunities For Improvement:**

1.	Please improve the annotations of the dataset. It is recommend to qualitatively and quantitively evaluate the quality of the annotation process.
2.	The design and results of experiments should consistent with the motivation and fully reflect your strengths (WIFI + RGB and LiDAR and mmWave). Why should we use WIFI besides LiDAR and mmWave?
3.	The visualization for WIFI and mmWave is not clearly presented, making it difficult to understand how it contributes to the HPE task.


**Relation To Prior Work:**

The content and writing of this paper are very similar to those of MRI[1]. This paper does not emphasize the differences between them clearly enough. Please consider providing an adequate comparative analysis of the work against previous studies to justify the novelty of their dataset.

**Summary And Contributions:**

This paper proposes a new multi-modal non-intrusive 4D human dataset to facilitate the algorithm development of wireless human sensing. It consists of four sensing modalities: RGB-D, LiDAR, mmWave radar, and WiFi. It composes 320.76k synchronized frames, and 27 categories of poses performed by 40 subjects. The collection process and the four data modalities are introduced, followed by experiments to produce a single-modal and multi-modality benchmark.

---

> ### Author Response · Authors · 2023-08-17
> **Response to Reviewer 2naR (part 1)**
>
> We thank the reviewer for many valuable suggestions. We worked hard on addressing these problems and revised the manuscript accordingly. We hope that your concerns can be addressed.
>
> > Q1. It is unclear about the quality of the dataset. This work uses 2D positions estimated by HRNet, and uses 2D positions estimated from two views to generate 3D positions. However, the correctness of the annotation process remains ambiguous. Please improve the annotations of the dataset. It is recommend to qualitatively and quantitatively evaluate the quality of the annotation process.
>
> **Answer**: As discussed in section 4.3 (Keypoint Quality), we have validated the quality of the data annotation by comparing a random subset with manual annotation. The reprojection error PCKh@0.5 is 95.66%, demonstrating the reliability of our annotation. We also show a video demo with annotations on our project page, and we have added the video into the supplementary materials to show the quality. Please note that we not only use the deep model HR-Net but also take the triangulation and tailored optimization techniques in Eq(2) and Eq(3) into consideration, as well as the specific refinement for actions containing occlusion cases. These skills that further improve the annotation quality have been described in section 4.3. It is seen that a similar annotation strategy is also applied to other dataset annotations [1][2]. In the future, we plan to extend the dataset to multi-user scenarios. To guarantee the annotation quality, we will utilize the Mocap system to acquire human motions for single and multiple users.
>
> > Q2. The design and results of experiments should be consistent with the motivation and fully reflect your strengths. Why should we use WIFI besides LiDAR and mmWave?
>
> **Answer**: Compared to mmWave radar and LiDAR, WiFi CSI has some unique advantages. Firstly, WiFi AP, as a ubiquitous infrastructure, has been widely deployed, while mmWave radar and LiDAR are expensive and need to be installed for applications. The future IEEE 802.11bf [1] will further bring WiFi sensing into more existing WiFi infrastructures. Secondly, mmWave radar is still restricted in some regions. For example, the widely-adopted 77-81GHz mmWave radars are not allowed to use in the indoor environment in Singapore, which limited the usage of mmWave radar. We believe that WiFi HPE will also enable some smart home applications in a cost-effective, and thus it is included in our dataset.
>
> > Q3. The visualization for WIFI and mmWave is not clearly presented, making it difficult to understand how it contributes to the HPE task.
>
> **Answer**: The visualization reflects that the HPE performances of WiFi-based and mmWave-based models are not good. We also added the video to show the pose estimation results of all modalities. Note that this is the performance of the baseline model in our benchmark. One of the important objectives of this dataset is to help build HPE models based on these unexplored sensors. We hope future research can improve the HPE performance using mmWave and WiFi by developing new techniques.

---

> ### Author Response · Authors · 2023-08-17
> **Response to Reviewer 2naR (part 2)**
>
> Q4. The motivation of the work is unclear. It is similar to mRI, which published in NeurIPS 2022 dataset track. The major differences are the introduction of the WIFI modality.
>
> **Answer**: The motivation of MM-Fi is to facilitate 4D human perception using wireless sensors, especially WiFi CSI and mmWave radar. Compared to mRI in NeurIPS-22 [1], the common point is that both datasets belong to HPE datasets. However, our MM-Fi dataset varies from mRI significantly in multiple aspects including sensor modality, action category, data scale, subject scale, multi-environment setting, and objectives. We have summarized these differences in the following table. Note that MM-Fi provides three different sensor modalities: WiFi, LiDAR (dense point cloud), and 60-64GHz mmWave radar. The radar used in mRI runs at 77-81GHz, which cannot be used in many regions due to the frequency control by the government. We argue that the motivation of our MM-Fi is strong, aiming at exploring wireless human sensing, and it has a larger data size and more sensor modalities, serving more sensing tasks and applications in the future.
>
> |  | mRI | MM-Fi |
> |---|---|---|
> | Sensor modality | IMU, RGBD, mmWave IWR1843 (77-81GHz) | RGBD, *mmWave IWR6843 (60-64GHz), WiFi, LiDAR* |
> | Action category | 13 | 27 |
> | Frames | 160k | 320.76k |
> | Subjects | 20 | 40 |
> | Environments | Single | Four distinct environments |
> | Objective | Rehabilitation | Wireless human sensing in privacy-preserving scenarios |
>
> > Q5. The design and results of multi-modal experiments do not fully reflect the advantages of WIFI.
>
> **Answer**: The advantage of WiFi is that WiFi has been widely adopted in buildings and private households, and it is more cost-effective than other modalities. One of the crucial reasons why we construct MM-Fi is to enable more accurate HPE using WiFi. The design and results of our experiments show the research gap of WiFi-based HPE. For the multi-modal fusion, we aim to provide more baselines to inspire the users. After more accurate WiFi-based HPE is achieved, we believe that its results can contribute to multi-modal fusion.
>
> > Q6. Several related HPE datasets are missing from the discussion, including mmbody, LIPD, LiCamPose, SLOPER4D, CIMI4D.
>
> **Answer**: We appreciate more related HPE datasets. We have read these papers carefully and added them to Table 1.
> mmbody [3] provides  3D Skeletons/Mesh dataset for mmwave radar and RGB(D) with 20 subjects, 100 activities in 6 different environments, containing more than 200k synchronized frames.
> Lidar-aid inertial pose [4] contains 15 performers, about 30 kinds of motions, and 62,341 frames of LiDAR point cloud and corresponding IMU measurements in total. It also provides ground-truth SMPL pose parameters and RGB images.
> LiCamPose [5] is a 3D-MPE dataset in long range wild scenes with a 128-beam OuSTER-1 LiDAR and a camera, with totally 8,980 frames of synchronized multimodal data. The ground truth is captured by Noitom Perception Neuron Studio(Noitom PN S). It contains 6 actions.
> Sloper4d [6] is an urban-level human pose dataset consisting of 15 sequences from 12 human subjects in 10 locations. There are a total of 100k LiDAR frames, 300k video frames, and 500k IMU-based motion frames captured over a total distance of more than 8 km and an area of up to 30,000$m^2$. It provides 3D mesh ground truth.
> CIMI4D [7] is a multi-modal climbing dataset that contains 60 minutes of RGB videos, 179,838 frames of LiDAR point clouds, 180 minutes of IMU poses, and an accurate global trajectory. It consists of 12 climbers to climb on 13 climbing walls, contained in 42 sequences contains. 2D and 3D keypoints are provided.
>
> **References**
>
> [1] mRI: Multi-modal 3d human pose estimation dataset using mmwave, RGB-d, and inertial sensors. In Thirty-sixth Conference on Neural Information Processing Systems Datasets and Benchmarks Track, 2022.
>
> [2] Humman: Multi-modal 4d human dataset for versatile sensing and modeling. In European Conference on Computer Vision (pp. 557-577). Cham: Springer Nature Switzerland.
>
> [3] mmbody benchmark: 3d body reconstruction dataset and analysis for millimeter wave radar. In Proceedings of the 30th ACM International Conference on Multimedia, pages 3501–3510, 2022.
>
> [4] Lidar-aid inertial poser: Large-scale human motion capture by sparse inertial and lidar sensors. arXiv preprint arXiv:2205.15410, 2022.
>
> [5] Weakly supervised 3d multi-person pose estimation for large-scale scenes based on monocular camera and single lidar. arXiv preprint arXiv:2211.16951, 2022.
>
> [6] Sloper4d: A scene-aware dataset for global 4d human pose estimation in 315 urban environments. In Proceedings of the IEEE/CVF Conference on Computer Vision and 316 Pattern Recognition, pages 682–692, 2023.
>
> [7] Cimi4d: A large multimodal climbing motion dataset under human-scene interactions. In Proceedings of the IEEE/CVF Conference on Computer Vision and Pattern Recognition, 463 pages 12977–12988, 2023.

---

> ### Author Response · Authors · 2023-08-17
> **Response to Reviewer 2naR (part 3)**
>
> >Q8. Typos and other questions:
> >Q8.1. In line 186, L_G and L_A should denote the general regularizer and action regularizer, respectively.
>
> **Answer**: We have revised this typo.
>
> > Q8.2. The description of the three settings (S2 and S3) in Data Splits is inconsistent with the experimental results. Should they be changed to the reverse order?
>
> **Answer**: Thanks for pointing out this issue. We wrote the Section 5.2 in a wrong order. We have revised the manuscript accordingly.
>
> >Q8.3. What’s the meaning of the expression “PA-MPJPE is much higher than MPJPE”?
>
> **Answer**: We are sorry for the incorrect claims here, and we have removed this sentence.
>
> > Q8.4. Is it unfair to use pretrained models for RGB, while retraining for LiDAR, mmWave, and WiFi in the baseline methods.
>
> **Answer**: We appreciate the suggestion and have trained the RGB model from scratch for all protocols and splits. We ran the experiments 3 times for each setting and reported the mean and std in the table below. We find that after training on MM-Fi, it achieves the best performance compared to other modalities. We will add the results to the final manuscript.
>
> |  | Protocol 1 | Protocol 1 | Protocol 2 | Protocol 2 | Protocol 3 | Protocol 3 |
> |---|---|---|---|---|---|---|
> | Settings | mpjpe | pa-mpjpe | mpjpe | pa-mpjpe | mpjpe | pa-mpjpe |
> | Random  | 64.70±0.51 | 34.75±0.06 | 68.52±1.39 | 31.46±0.08 | 60.54±0.36 | 32.45±0.18 |
> | Environment | 104.75±0.73 | 40.35±0.14 | 95.72±1.09 | 38.75±0.10 | 97.96±0.89 | 39.18±0.06 |
> | Subject | 88.64±1.06 | 35.80±0.09 | 82.81±1.48 | 32.08±0.06 | 85.67±0.47 | 33.44±0.05 |
>
> > Q8.5. In lines 297 and 298 of the paper, the fusion of RGB images and LiDAR (I+L) outperforms LiDAR significantly for MPJPE under three settings. However, from the experimental table, LiDAR is better than the fusion of RGB images and LiDAR (I+L) under S3.
>
> **Answer**: We think the reviewer may happen to read the wrong results. As shown in Table 3 and 4, the I+L achieves the MPJPE of 159.6, 153.5, and 192.1 under S3 for three protocols, while the single LiDAR model achieves the MPJPE of 192.3, 186.0, 303.8. The MPJPE is the error, so the less it is, the better HPE performance we get. Hence, the fusion model (I+l) results are much better than the single LiDAR model.

---

> > ### Comment · Reviewer_2naR · 2023-08-17
> >
> > Thanks for the response from the authors. The authors address my concerns. Thus, I have upgraded the rating from 5 to 7.

---

> > > ### Author Response · Authors · 2023-08-17
> > > **Thanks for the review and increasing rating**
> > >
> > > Thanks for your prompt responses. We are happy that your concerns have been addressed and appreciate that the rating has been increased.

---

### Official Review · Reviewer_j8DT · 2023-07-18
**Generally Good Work for Multi-Modal Human Sensing**

**Rating:** 6
**Confidence:** 4
**Correctness:** Yes. The claims made in the submissio…
**Clarity:** Yes, the paper is well written.

**Strengths:**

1. This paper collects a real-world dataset, where the actions of individuals are monitored by different sensors, including RGB cameras, depth cameras, LiDAR, mmWave, and WiFi. It is a pioneering dataset of multi-modal human sensing.
2. The data collection process, involving signal gathering, synchronization, and annotation, is well explained.
3. Good analysis and data visualization are provided.
4. Using different protocols and settings, the usefulness of the dataset is well illustrated.


**Additional Feedback:**

No additional feedbacks.

**Documentation:**

It seems to be well documented.

**Ethics:**

No ethics issues.

**Limitations:**

1. Though this dataset has already included 27 actions, some representative daily actions seem to be neglected, such as sitting down, standing up, etc.
2. The authors used the number of frames (320k) to measure data volume, which I think is not quite intuitive, since it may be related to different frame rates, etc. I think using the time span of data (e.g., 5 or 10 hours) may be more appropriate.
3. Line 145 says the CSI frames are 3$\times$114$\times$T, and T=32ms. However, I notice that the frames in the dataset are 3$\times$114$\times$10. The authors may need to clarify the reasons for this.
4. In the dataset, there are some "-inf" values in some sequences. The authors may need to clarify the reason and the solutions.
5. Finally, the authors can provide more detailed future work for the better exploitation of the dataset.

**Opportunities For Improvement:**

1. Only 1080 sequences are collected in the dataset, which can be enlarged for more representative data, helping models better understand human features in the data.
2. In each sequence, the individuals were repeating their actions. It seems that the authors can further segment each sequence to have more fine-grained samples.


**Relation To Prior Work:**

Yes. The difference of this work from previous works has been discussed.

**Summary And Contributions:**

This paper proposes MM-Fi, a 4D human sensing dataset including multiple modalities. It consists of 27 human actions from 40 individuals, recording 5 data modalities. The authors performed experiments to evaluate the performance of human pose estimation, regarding different modalities. Generally, this dataset covers various scenarios and is beneficial to future studies.

---

> ### Author Response · Authors · 2023-08-17
> **Response to Reviewer j8DT (part 1)**
>
> We thank the reviewer for the appreciation of our work and many valuable suggestions. We have answered all the questions and revised the manuscript accordingly. We hope that these concerns can be addressed.
>
> > Q1. Only 1080 sequences are collected in the dataset, which can be enlarged for more representative data, helping models better understand human features in the data.
>
> **Answer**: We agree with the reviewer that more sequences can help models better understand human features. Our current setting is quite simple, and the objective is to enable the community to explore more sensor modalities and multi-modal fusion for human sensing. Compared to the existing wireless multi-modal datasets in Table 1, e.g., MARS and mRI, our number of sequences has been much larger than theirs. To achieve very robust HPE based on these modalities, we expect to include more settings that will enlarge the MM-Fi 2.0 by introducing multi-persons, multi-locations, and multi-orientations. We have added the limitation and the future work in the manuscript.
>
> > Q2. In each sequence, the individuals were repeating their actions. It seems that the authors can further segment each sequence to have more fine-grained samples.
>
> **Answer**: We thank the reviewer for the suggestion and have begun segmenting the sequences to have more fine-grained annotations. As stated in the revised Section 4.3 Page 7 Lines 214-220, with the details of the segmentation process as presented in the Appendix (Appendix Page 3, Lines 55-66). The segmentation process is performed by human annotators with an automated segment program that enables the generation of segments through human selected breakpoints (frame). To ensure correct annotation, a voting process is performed where the breakpoints that are selected by the majority of the annotators are viewed as the ground-truth breakpoints. The segments are frames between each breakpoint. The resulting fine-grained samples are recorded in a .csv file with the sample records presented as in Table 7 in the Appendix (Appendix Page 3). The .csv file includes the Environment, Subject, Action, and Segments information, where the different segments are represented by the beginning and end frames. We have finished the manual annotation and added this file (action_segmentation_annotation.csv) to the supplementary material.
>
> > Q3. Though this dataset has already included 27 actions, some representative daily actions seem to be neglected, such as sitting down, standing up, etc.
>
> **Answer**: At present, we design the 27 actions by taking into account the changes in the joints of the whole body as much as possible, so that the human pose estimation can cover most of the body joints. Compared to the current multimodal datasets in Table 1, we already have many actions. We agree that enlarging the number of actions will be very helpful. In MM-Fi 2.0, we plan to extend the number of actions, scenes, persons, and orientations. We have added these to the limitation sections in the manuscript.
>
> > Q4. The authors used the number of frames (320k) to measure data volume, which I think is not quite intuitive. I think using the time span of data (e.g., 5 or 10 hours) may be more appropriate.
>
> **Answer**: We appreciate your suggestions. For human pose estimation (HPE) datasets, most of them actually use the number of frames or sequences, while very few datasets provide the time span of data, as shown in the HPE survey [5]. That’s why we provided the number of frames and sequences in Table 1. However, we find that some of the datasets provide the sampling rate. Thus, we calculate the time span using the sampling rate and the number of frames for some multi-modal datasets in Table 1 if they are available. As shown in the table below, our MM-Fi still has a time span of 9 hours in total, greater than many existing datasets. The Waymo open dataset comes from the autonomous driving scene so it serves different outdoor applications. We hope these results are more intuitive to make comparisons.
>
> | Dataset  | mRI [1]    | WiPose [2] | Waymo [3]   | HPTE [4] | MM-Fi (ours) |
> |----------|------------|------------|-------------|----------|--------------|
> | Duration | 4.44 hours | 2.67 hours | 10.83 hours | 2.00 hours  | 9.00 hours      |

---

> ### Author Response · Authors · 2023-08-17
> **Response to Reviewer j8DT (part 2)**
>
> > Q5. Line 145 says the CSI frames are 3x114xT, and T=32ms. However, I notice that the frames in the dataset are 3x114x10. The authors may need to clarify the reasons for this.
>
> **Answer**: Sorry for the typo and confusion caused by the wrong description. In fact, $T$ is the number of original frames in each CSI matrix during a period of 100ms, instead of the time period. $T=10$ and we have the CSI matrix of 3x114x10 for each timestamp. We have corrected this typo and revised the description in the manuscript, which can be referred to in Section 3.1 WiFi CSI Data.
>
> > Q6. In the dataset, there are some "-inf" values in some sequences. The authors may need to clarify the reason and the solutions.
>
> **Answer**: The “-inf” number comes from the noise or empty frames from the CSI tool. In our benchmark, we deal with these numbers by linear interpolation. To facilitate the users, we have embedded these processing codes into our dataset tool. When the user loads our WiFi CSI data, these numbers will be handled by linear interpolation. The codes can be found in the following link. (https://github.com/ybhbingo/MMFi_dataset/blob/c58e8746a6645ec7c37dba19c0c10c20530cf89e/mmfi_lib/mmfi.py#L233-L245)
> We preserve these “-inf” numbers since the users may want to handle them with better techniques in the future. We have added this information into our appendix.
>
> > Q7. Finally, the authors can provide more detailed future work for the better exploitation of the dataset.
>
> **Answer**: We agree that the future work should be illustrated in detail. To this end, we add one subsection to introduce the current limitation and how to extend the MM-Fi 2.0 in Section.6, and one subsection to illustrate what future research could be enabled in Appendix.A.9. We attach the contents in these subsections into the next rebuttal box for your convenience.
>
> **References**
>
> [1] An, S., Li, Y., & Ogras, U. (2022). mri: Multi-modal 3d human pose estimation dataset using mmwave, rgb-d, and inertial sensors. Advances in Neural Information Processing Systems, 35, 27414-27426.
>
> [2] Jiang, W., Xue, H., Miao, C., Wang, S., Lin, S., Tian, C., ... & Su, L. (2020, April). Towards 3D human pose construction using WiFi. In Proceedings of the 26th Annual International Conference on Mobile Computing and Networking (pp. 1-14).
>
> [3] Zheng, J., Shi, X., Gorban, A., Mao, J., Song, Y., Qi, C. R., ... & Anguelov, D. (2022). Multi-modal 3d human pose estimation with 2d weak supervision in autonomous driving. In Proceedings of the IEEE/CVF Conference on Computer Vision and Pattern Recognition (pp. 4478-4487).
>
> [4] Ar, I., & Akgul, Y. S. (2014). A computerized recognition system for the home-based physiotherapy exercises using an RGBD camera. IEEE transactions on neural systems and rehabilitation engineering : a publication of the IEEE Engineering in Medicine and Biology Society, 22(6), 1160–1171. https://doi.org/10.1109/TNSRE.2014.2326254
>
> [5] Zheng, C., Wu, W., Chen, C., Yang, T., Zhu, S., Shen, J., ... & Shah, M. (2020). Deep learning-based human pose estimation: A survey. ACM Computing Surveys.

---

> ### Author Response · Authors · 2023-08-17
> **Response to Reviewer j8DT (part 3)**
>
> > These contents are what we added for question 7.
>
> #### Section.6 Limitation
>
> The MM-Fi dataset currently has limitations regarding annotations and benchmarks. Firstly, the annotation process is manual and the quality is limited. Due to the resolution of wireless sensing data, the current sensing tasks are restricted to activity level and keypoint level, which have been validated with high quality. However, for tasks that require higher resolution, such as dense pose estimation, we only provide annotations obtained from algorithms that have not been validated. These annotations are included to facilitate new tasks and inspire further research. In the upcoming MM-Fi V2.0, we plan to use a motion capture system to annotate the dense pose to address this limitation. Secondly, as the first dataset to offer mmWave radar, LiDAR, RGBD, and WiFi data simultaneously, there are tasks that have not been extensively studied yet. Therefore, some baseline methods developed by us may not perform optimally without careful design. Our intention is to inspire researchers to explore these unexplored fields and contribute to the future benchmarking of the MM-Fi dataset. Thirdly, the current dataset is collected in a controlled condition, i.e., 3m away and same facing direction, with a single person. We plan to include multi-orientation, multi-location, and multi-user scenarios in MM-Fi 2.0. We acknowledge these limitations and aim to improve the dataset by addressing them in future versions. Our objective is to provide a comprehensive resource that encourages research and advancements in the non-intrusive wireless sensing research community.
>
> #### A.9 Future Research using MM-Fi
>
> MM-Fi significantly expands the horizons of research by furnishing meticulously synchronized multimodal human sensing data. As advised by the reviewer, we have encapsulated these novel research directions as follows, and have incorporated these enhancements into the manuscript:
>
> 1. Cross-domain wireless sensing. The realm of wireless sensing is often challenged by recognition performance disparities stemming from domain shifts, which arise due to variations in environments and subjects. While this issue has garnered extensive exploration within action recognition based on RGB data, it remains relatively uncharted within the context of Human Pose Estimation (HPE) utilizing modalities such as mmWave, LiDAR, and WiFi. The MM-Fi dataset presents a unique opportunity to address this gap by facilitating research into domain adaptation and generalization for HPE on these emerging modalities. By enabling investigations into the adaptation of models across diverse domains, MM-Fi opens avenues for enhancing the robustness and applicability of wireless sensing technologies.
> 2. Cross-modal supervision for fine-grained wireless sensing. Previous works have showcased the utility of WiFi and mmWave radar for action recognition tasks. However, MM-Fi introduces a transformative dimension by offering meticulously synchronized multimodal data and comprehensive annotations. Based on MM-Fi, cross-modal learning can enable these sparse data modalities to achieve fine-grained recognition tasks, e.g., human pose estimation and action segmentation.
> 3. Multi-modal wireless sensing. MM-Fi's integration of five distinct modalities empowers researchers to explore the potential of multi-modal wireless sensing, where different sensing technologies complement each other to achieve more comprehensive and accurate insights into human behavior. For example, integrating RGB and WiFi can overcome the illumination issue of RGB-based solutions. This dataset serves as a launchpad for pioneering investigations into techniques that fuse information from LiDAR, mmWave radar, and WiFi signals to attain a holistic understanding of human actions and interactions. As researchers delve into multi-modal fusion methods, the MM-Fi dataset becomes a valuable resource for the development of advanced solutions that leverage the strengths of each modality, while compensating for their individual limitations.

---

> ### Author Response · Authors · 2023-08-23
> **Looking forward to you reply**
>
> Dear Reviewer,
>
> We have answered the question and submitted the revised manuscript as well as supplementary materials. We are looking forward to your reply.
>
> Feel free to let us know if you have any other concerns. Thanks!
>
> Best Regards,
>
> Authors of Submission 44

---

> ### Author Response · Authors · 2023-08-27
> **Update: The manual annotation of action segments has been finished and updated**
>
> Dear Reviewer,
>
> We sincerely appreciate your constructive suggestions.
>
> We have just finished the annotation process mentioned in Q2, and added this new annotation file (action_segmentation_annotation.csv) to the supplementary material. The .csv file includes the Environment, Subject, Action, and Segments information, where the different segments are represented by the beginning and end frames. We hope this annotation can facilitate action detection and recognition tasks in the future.
>
> Feel free to let us know if you have any other advice. Thanks!
>
> Best regards,
> Authors of Submission 44

---

> > ### Comment · Reviewer_j8DT · 2023-08-30
> > **Generally Good Work for Multi-Modal Human Sensing**
> >
> > Dear authors,
> >
> > I greatly appreciate your responses and hard work on the paper revision. Most of my concerns have been resolved. Thank you so much.
> >
> > Best wishes,
> >
> > Reviewer j8DT

---

> ### Comment · Area_Chair_tqoB · 2023-08-29
> **Any comment from the reviewer?**
>
> Dear Reviewer j8DT ,
>
> Your active participation is vital.
> Please check the authors' responses thoroughly, revisit your reviews in light of the authors' rebuttals, engage in discussions to resolve conflicts, and remember the importance of timely responses.
>
> Thank you for your continued dedication.

---

### Official Review · Reviewer_bm2n · 2023-07-20
**Good multimodal dataset, More details of experiment environments needed, layout can be more diverse**

**Rating:** 6
**Confidence:** 4
**Clarity:** Yes, it is well written!

**Strengths:**

(1) The dataset captures 27 different types of activities, which is pretty diverse.

(2) The dataset can be used for activity recognition, 3D pose estimation, domain adaptation, etc. It is applicable to a number of research questions.

(3) The authors demonstrated good effort toward sensing platform building and synchronization. The paper is generally well-written.

**Additional Feedback:**

In equation 2, what are p_n and k_n?

It would also be interesting to see how SOTA models perform on this dataset when the task is activity recognition.

**Correctness:**

The paper is generally making reasonable claims. I only have some questions about its ground truth:

(6) After reading, my understanding is that the depth camera actually serves as the ground truth sensor here -- the human pose is estimated using a learning-based model running on IR camera data. However, the IR camera has only the 3D front side of the user and can suffer from the user's self-occlusions. Would adopting a motion capture system be better for ground truth collection?

If we use a neural model to predict the pose from a sensor, then develop other models using other sensors to approximate the 'ground truth,' then these developed models can only be worse or, ideally, as good as the ground truth model. Thus logically, it is critical to prove that the ground truth model (D435 data + HRNet-48) is good enough.

**Documentation:**

It is well documented.

**Ethics:**

IRB approval explicitly mentioned.

**Limitations:**

(4) With one more page allowed in camera ready, it is necessary to have a "limitation" session honestly describing what is not yet there in this work.

(5) For mmWave, what is the intuition of using a processed point cloud instead of an azimuth heatmap and/or range dopper? The pointcloud is a "second-hand" information obtained by processing the raw features using algorithms like CFAR. The authors also mentioned that the pointcloud from mmWave is extremely sparse. Predicting human pose from such a sparse pointcloud may be an ill-defined problem as the pointcloud is not informative enough -- the 'pose' we see in the end could just be fingerprinting or template matching results. Yes, aggregating the points over time may get us more points, but we pay the price of time resolution.

**Opportunities For Improvement:**

(1) The paper did not talk much about the environment. From the evaluations, there seem to be four environments involved. I think it is necessary to provide some details. Sensing modalities, especially CSI, can easily overfit to a particular environment settings and cause models developed with such datasets to fail to generalize.

(2) Following (1), if my understanding is correct, the dataset captures users performing daily activities when users stand in front of the sensors (in the line-of-sight of WiFi). Are the changes of user positions taken into consideration? For wireless sensors, the signal with users standing in the line-of-sight and the signal where users stand in Fresnel Zones can be very different.

(3) How the alignment of frames is done is not clear. Line 145 states that CSI frames are 32ms/frame, Line 135 states mmWave point cloud is aggregated at 2Hz, and videos are typically 30/60Hz. How are the frames got aligned? The raw sensor data may have timestamps with them, but it is important to talk about frame alignment as well.



**Relation To Prior Work:**

Compared to previous datasets, this one has more modalities, more activities, and an interesting focus on activity pose estimations.

**Summary And Contributions:**

This paper proposes a multimodal dataset consisting of camera, depth camera, mmWave radar, commercial LiDAR, and WiFi CSI. It captures users performing daily activities when users stand in front of the sensors (in the line-of-sight of WiFi). The dataset is mainly proposed for 3D skeleton reconstruction.

---

> ### Author Response · Authors · 2023-08-17
> **Response to Reviewer bm2n (part 1)**
>
> We thank the reviewer for the appreciation of our work and many valuable suggestions. We have answered all the questions and revised the manuscript accordingly. We hope that these concerns can be addressed.
>
> > Q1. The paper did not talk much about the environment. From the evaluations, there seem to be four environments involved. I think it is necessary to provide some details. Sensing modalities, especially CSI, can easily overfit to a particular environment settings and cause models developed with such datasets to fail to generalize.
>
> **Answer**: The MM-Fi consists of four different environments (i.e., the different settings in two labs), and we have added the layouts of these layout settings in the appendix. We agree that overfitting can happen for some sensing modalities, and therefore in the benchmark settings, we design S2 (cross-subject) and S3 (cross-environment) evaluation protocols to examine the generalization ability of the model for specific modalities. We can see from the results that WiFi CSI severely suffers from the domain issue. We believe that our dataset contributes to addressing the cross-domain issue for WiFi sensing. We have added one paragraph in the appendix to illustrate how the dataset contributes to the future research direction.
>
> > Q2. Following (1), if my understanding is correct, the dataset captures users performing daily activities when users stand in front of the sensors (in the line-of-sight of WiFi). Are the changes of user positions taken into consideration? For wireless sensors, the signal with users standing in the line-of-sight and the signal where users stand in Fresnel Zones can be very different.
>
> **Answer**: Yes, we have considered the changes in user position. The designed actions can be roughly divided into two classes depending on whether the user needs whole-body movements. For actions (e.g. A07/A08) that do not contain leg movement, the center of the user’s body mass is kept near the LoS between WiFi Tx and Rx, which means the user motions are better sensed by the WiFi CSI. For other actions, (e.g. A09/A10) which need periodical body movements, the center of the user’s body mass is no longer kept fixed near the LoS but still within the first tenth Fresnel Zone (ellipsoid with a radius of about 0.735m in our setting), which also guarantees the sensibility of WiFi CSI because the first ten FFZs dominate the power of WiFi signal transmission. To demonstrate the difference between the WiFi signals on LoS and n-th Fresnel Zone, we supplement the results (in the next box) for each human action. We can observe from the benchmark results that, the HPE model performs better on A07/A08 (limb extensions) than A09/A10 (lunges towards left-front/right-front direction) and A15/A16 (lunges towards left/right direction), in which the user’s body is not in the LoS of WiFi.
>
> > Q3. How the alignment of frames is done is not clear. Line 145 states that CSI frames are 32ms/frame, Line 135 states mmWave point cloud is aggregated at 2Hz, and videos are typically 30/60Hz. How are the frames got aligned? The raw sensor data may have timestamps with them, but it is important to talk about frame alignment as well.
>
> **Answer**: Sorry for the confusion caused by the unclear description. It is true that the WiFi, mmWave radar, LiDAR, and RGB have different sampling rates. To deal with the different sampling rates, we align the modality frames in ROS according to their timestamps. As discussed in Section 3.2, all the sensor modalities are saved in a ROS bag with the timestamp at each frame. According to the sampling timestamp, we set a 10Hz sampling timestamp and retrieve the multi-modal data frames that are closest to this timestamp. In this manner, we can retrieve 10Hz multi-modal data in ROS and the synchronization error is less than 25ms for all frames and modalities. The error is calculated according to the lowest sampling rate of sensors, i.e., 20Hz. The subsequent aggregation of mmWave point cloud is actually a post-processing method to make the data less sparse. We have added these contents to the manuscript.

---

> ### Author Response · Authors · 2023-08-17
> **Response to Reviewer bm2n (part 2)**
>
> | Modality  | RGB   | RGB      | LiDAR | LiDAR    | mmWave | mmWave   | WiFi   | WiFi     |
> |-----------|-------|----------|-------|----------|--------|----------|--------|----------|
> | Criterion | MPJPE | PA-MPJPE | MPJPE | PA-MPJPE | MPJPE  | PA-MPJPE | MPJPE  | PA-MPJPE |
> | A01       | 82.5  | 31.2     |  91.3 |   60.0   |  151.1 |   65.0   | 179.4  | 102.1    |
> | A02       | 54.8  | 28.6     |  75.3 |   60.9   |  82.2  |   40.9   | 154.5  | 119.4    |
> | A03       | 56.1  | 37.8     |  81.4 |   62.4   |  85.8  |   49.7   | 159.6  | 141.1    |
> | A04       | 64.0  | 35.7     | 105.2 |   68.7   |  86.7  |   52.2   | 165.6  | 103.9    |
> | A05       | 57.8  | 36.1     | 102.6 |   67.8   |  87.3  |   48.5   | 166.3  | 108.7    |
> | A06       | 65.0  | 42.0     |  98.0 |   76.1   |  79.4  |   47.1   | 234.9  | 121.5    |
> | A07       | 40.7  | 23.1     |  64.8 |   46.4   |  132.4 |   48.6   | 145.0  | 101.2    |
> | A08       | 39.8  | 25.4     |  69.0 |   50.4   |  193.3 |   51.6   | 135.6  | 103.1    |
> | A09       | 98.6  | 39.9     | 111.3 |   88.8   |  116.8 |   65.6   | 263.4  | 157.6    |
> | A10       | 72.4  | 36.7     |  98.7 |   75.3   |  117.6 |   63.2   | 244.7  | 152.0    |
> | A11       | 43.5  | 19.2     |  67.2 |   45.9   |  136.2 |   48.7   | 159.8  | 112.6    |
> | A12       | 58.7  | 31.6     |  85.5 |   60.4   |  102.3 |   55.1   | 211.4  | 120.5    |
> | A13       | 36.1  | 22.6     |  68.2 |   47.0   |  135.4 |   57.2   | 151.0  | 115.5    |
> | A14       | 49.6  | 31.6     |  71.1 |   48.5   |  139.3 |   51.2   | 138.0  | 115.8    |
> | A15       | 57.3  | 27.3     |  83.4 |   50.2   |  102.1 |   49.9   | 256.7  | 131.0    |
> | A16       | 58.0  | 26.3     |  87.3 |   50.7   |  100.9 |   46.8   | 252.9  | 132.0    |
> | A17       | 52.0  | 28.4     |  67.5 |   48.0   |  206.6 |   73.1   | 177.6  | 142.2    |
> | A18       | 48.6  | 28.8     |  72.1 |   50.8   |  265.8 |   76.5   | 165.5  | 146.7    |
> | A19       | 83.7  | 51.2     | 112.8 |   76.6   |  113.7 |   65.4   | 281.0  | 135.6    |
> | A20       | 71.2  | 40.9     | 100.7 |   69.5   |  96.2  |   59.9   | 203.9  | 120.7    |
> | A21       | 62.9  | 40.6     | 101.1 |   67.4   |  100.0 |   58.7   | 212.0  | 119.5    |
> | A22       | 73.3  | 38.8     |  92.7 |   64.2   |  98.3  |   51.3   | 199.6  | 112.0    |
> | A23       | 76.5  | 42.2     |  99.1 |   67.6   |  98.7  |   54.0   | 205.8  | 108.0    |
> | A24       | 48.5  | 25.9     |  80.2 |   47.2   |  139.1 |   47.9   | 170.3  | 100.8    |
> | A25       | 41.3  | 26.6     |  81.0 |   49.8   |  114.6 |   49.9   | 173.1  | 100.0    |
> | A26       | 88.2  | 30.8     |  94.8 |   61.9   |  93.1  |   55.3   | 307.5  | 156.2    |
> | A27       | 64.5  | 32.6     |  89.9 |   51.7   |  80.0  |   38.9   | 192.1  | 106.4    |

---

> ### Author Response · Authors · 2023-08-17
> **Response to Reviewer bm2n (part 3)**
>
> > Q4. With one more page allowed in camera ready, it is necessary to have a "limitation" session honestly describing what is not yet there in this work.
>
> **Answer**: We agree with the reviewer’s suggestion. Actually, we already included a limitation session in the appendix (A.7). We will summarize more limitations pointed out in the review process, and add this section into the manuscript. The limitation section is shown as follows:
>
> The MM-Fi dataset currently has limitations regarding annotations and benchmarks. Firstly, the annotation process is manual and the quality is limited. Due to the resolution of wireless sensing data, the current sensing tasks are restricted to activity level and keypoint level, which have been validated with high quality. However, for tasks that require higher resolution, such as dense pose estimation, we only provide annotations obtained from algorithms that have not been validated. These annotations are included to facilitate new tasks and inspire further research. In the upcoming MM-Fi V2.0, we plan to use a motion capture system to annotate the dense pose to address this limitation. Secondly, as the first dataset to offer mmWave radar, LiDAR, RGBD, and WiFi data simultaneously, there are tasks that have not been extensively studied yet. Therefore, some baseline methods developed by us may not perform optimally without careful design. Our intention is to inspire researchers to explore these unexplored fields and contribute to the future benchmarking of the MM-Fi dataset. Thirdly, the current dataset is collected in a controlled condition, i.e., 3m away and same facing direction, with a single person. We plan to include multi-orientation, multi-location, and multi-user scenarios in MM-Fi 2.0. We acknowledge these limitations and aim to improve the dataset by addressing them in future versions. Our objective is to provide a comprehensive resource that encourages research and advancements in the non-intrusive wireless sensing research community.
>
> > Q5. For mmWave, what is the intuition of using a processed point cloud instead of an azimuth heatmap and/or range dopper? The pointcloud is a "second-hand" information obtained by processing the raw features using algorithms like CFAR. Also, aggregating the points over time may get us more points, but we pay the price of time resolution.
>
> **Answer**: We agree that preserving the raw data should be better. However, this has not been achieved yet due to the limitation of the radar SDK provided by TI. As we used ROS on Linux to synchronize all modalities, we tried to extract raw data (or range Doppler) in ROS before we built the sensor platform. We used DCA1000 integrated with IWR6843 to extract raw data using the SDK provided by TI,  but we found that the SDK only supported Windows with a TI software. We hope that TI can develop more SDK in the future so that we can integrate it into our sensor platform. For the data aggregation, we agree with the reviewer. We will also release the raw data without aggregation in the next version.
>
> > Q6. Would adopting a motion capture system be better for ground truth collection? If we use a neural model to predict the pose from a sensor, it is critical to prove that the 'ground truth' model (D435 data + HRNet-48) is good enough.
>
> **Answer**: We appreciate the suggestions on capture systems. We did not use full-body motion capture (Mocap) systems due to several reasons. Firstly, one of the crucial features of MM-Fi is to provide data from multiple environments for cross-scene wireless human pose estimation, but the deployment of a Mocap system requires the devices to be installed at relatively fixed locations. Moving the entire equipment is very inconvenient, e.g., OptiTrack and Vicon. Secondly, these systems are of high cost, while our low-cost annotation solution also provides accurate annotations validated by manual annotation.
>
> We have validated the quality of the data annotation by comparing a random subset with manual annotation in section 4.3. The reprojection error PCKh@0.5 is 95.66%, demonstrating the reliability of our annotation. Note that we not only use deep learning algorithms, but also take the triangulation and optimization in Eq(2) and Eq(3) into consideration, as well as the specific refinement for actions containing occlusion cases.  We notice that such an annotation strategy is also applied to other dataset annotations [1][2]. In the future, we plan to extend the dataset to multi-user scenarios. To guarantee the quality, we will install the Mocap system in the MM-Fi 2.0.
>
> >Q7: In equation 2, what are p_n and k_n?
>
> **Answer**: As illustrated in Line-184, the $p_n$ represents the 3D keypoints and $k_n$ indicates the 2D keypoints.
>
> >Q8: It would also be interesting to see how SOTA models perform on this dataset when the task is activity recognition.
>
> **Answer**: Thanks for the suggestions. We have added the action recognition benchmark in the appendix.

---

> ### Author Response · Authors · 2023-08-17
> **Response to Reviewer bm2n (part 4)**
>
> **References**
>
> [1] Sizhe An, Yin Li, and Umit Ogras. mRI: Multi-modal 3d human pose estimation dataset using mmwave, RGB-d, and inertial sensors. In Thirty-sixth Conference on Neural Information Processing Systems Datasets and Benchmarks Track, 2022.
>
> [2] Cai, Z., Ren, D., Zeng, A., Lin, Z., Yu, T., Wang, W., ... & Liu, Z. (2022, October). Humman: Multi-modal 4d human dataset for versatile sensing and modeling. In European Conference on Computer Vision (pp. 557-577). Cham: Springer Nature Switzerland.

---

> ### Author Response · Authors · 2023-08-23
> **Looking forward to your reply**
>
> Dear Reviewer,
>
> We have answered the question and submitted the revised manuscript as well as supplementary materials. We are looking forward to your reply.
>
> Feel free to let us know if you have any other concerns. Thanks!
>
> Best Regards,
>
> Authors of Submission 44

---

> > ### Comment · Reviewer_bm2n · 2023-08-24
> >
> > Dear authors,
> >
> > Firstly, I would like to thank you for the considerable effort you have put into improving the quality of this paper. Most of my concerns were addressed.
> >
> > I would like to comment on Q5. I understand that getting the raw I/Q data is hard since DCA1000 is necessary. I was talking about information like the azimuth heatmap and range-Doppler heatmap, which is possible to obtain using ROS. You might want to check "https://github.com/m6c7l/pymmw" for future reference.
> >
> > Also, I have a general comment on Q1-2. Given WiFi's low resolution, I had some doubts about the series of WiFi-CSI-based human pose estimation work; it is likely that the neural networks are memorizing/fingerprinting the correspondence between CSI data and human gestures. If so, such a model will have very limited deployment value since it does not generalize well to unseen environments/conditions. To verify if this is really the case, I always hoped there is a dataset containing many different environment layouts and sensor placements, and the target may stand at many places across the room while the WiFi Tx-Rx remains stable. So, I am looking forward to an MM-Fi 2.0 dataset.
> >
> > I would like to keep my decision that this paper is a marginal accept to me. I see some merits and considerable effort in this paper.
> >
> > Thanks!

---

> > > ### Author Response · Authors · 2023-08-25
> > >
> > > Dear reviewer,
> > >
> > > We appreciate the response and more constructive suggestions. We will include your valuable suggestions in the future MM-Fi 2.0 dataset:
> > > - For Q5, we will definitely embed pymmw in ROS, which could preserve the azimuth heatmap and range-Doppler heatmap.
> > > - For Q1-2, we agree that the multi-location multi-angle data is very useful for improving the generalization ability of deep models on WiFi human pose estimation (HPE), which will be incorporated in MM-Fi 2.0.
> > >
> > > We hope our MM-Fi work will contribute to multi-modal HPE and wireless HPE tasks.
> > >
> > > Thanks!

---

### Official Review · Reviewer_boR4 · 2023-07-21
**MM-Fi: Multi-Modal Non-Intrusive 4D Human Dataset for Versatile Wireless Sensing**

**Rating:** 6
**Confidence:** 3
**Correctness:** Yes.
**Clarity:** Yes, well organized.

**Strengths:**

- Provide the first pose estimation dataset that comprises various sensing modalities (RGB-D, LiDAR, radar, and WiFi) and construct a platform for synchronized collections.
- Advantages in terms of diversities compared to previous datasets: more subjects and diverse action classes.
- Provide benchmarks on multi-modal human pose estimation and explore its potentiality numerically.
- The dataset is well documented.

**Additional Feedback:**

None.

**Documentation:**

Yes. It would be beneficial for providing detailed specifications of each sensor (mmWave radar, for example, # Tx/Rx, range resolution, ...).

**Ethics:**

No.

**Limitations:**

- The authors provide the limitations of work in terms of annotation and fusion benchmark. In my opinion, there are also limitations in terms of movement diversity (i.e., each subject is fixed in a certain location and facing angle).

**Opportunities For Improvement:**

- What is the motivation for using WiFI CSI in pose estimation, compared to mmWave signals?
- It seems that the collections are biased towards (3m, 0 degree) while always facing towards the sensor, which are quite unrealistic. How would be the performance when a person becomes far away, locates in different azimuth angles, or has a global movements (i.e., changes in locations, orientations, …)
- ‘Resolution’ in Table 2 arises confusions. For example,  mmWave has superior resolutions in depth, while maintaining resolutions in lateral dimensions. Does this ‘Resolution’ represent angular resolutions? Does ‘WiFi’ have generally poor resolutions in all dimensions?
- Considering the poor performance of WiFi, it is doubt that WiFi would provide additional gain when fused with other modalities.
- S2, S3 demonstrated in L239—L240 and those explained in Section 5.2 are different.

**Relation To Prior Work:**

Yes.

**Summary And Contributions:**

* Propose a non-intrusive, multi-modal 4D pose estimation dataset, composed of RGB-D, LiDAR, mmWave radar, and WiFI signals, and provide single-modal and multi-modal benchmark.

---

> ### Author Response · Authors · 2023-08-17
> **Response to Reviewer boR4 (part 1)**
>
> We thank the reviewer for the appreciation of our work and many valuable suggestions. We have answered all the questions and revised the manuscript accordingly. We hope that these concerns can be addressed.
>
> > Q1. What is the motivation for using WiFI CSI in pose estimation, compared to mmWave signals?
>
> **Answer**: We admit that mmWave radar has higher accuracy than WiFi for human pose estimation (HPE) tasks. Nevertheless, compared to mmWave radar, WiFi CSI has some advantages for pose estimation and action recognition. Firstly, WiFi AP, as a ubiquitous infrastructure, has been widely deployed, while mmWave radar device still requires to be bought and installed. The future IEEE 802.11bf [1] will further bring WiFi sensing into more existing WiFi infrastructures. Secondly, mmWave radar is still restricted in some regions. For example, the widely-adopted 77-81GHz radars are not allowed to use in the indoor environment in Singapore, which limited the usage of some mmWave radar. We believe that WiFi HPE will also enable some smart home applications, and thus it is included in our dataset.
>
> [1] Du, R., Xie, H., Hu, M., Xin, Y., McCann, S., Montemurro, M., ... & Xu, J. (2022). An overview on IEEE 802.11 bf: WLAN sensing. arXiv preprint arXiv:2207.04859.
>
> > Q2. It seems that the collections are biased towards (3m, 0 degree) while always facing towards the sensor. How would be the performance when a person becomes far away or has a global movements?
>
> **Answer**: We admit that the current MM-Fi dataset is collected in a controlled condition (3m, fixed direction). It serves for applications where the person can be required to perform activities in a specific location, e.g., sports rehabilitation and metaverse gamings. From the benchmark results, we observe that even under the simple condition, the WiFi-based and mmWave radar based HPE still generate unsatisfactory results, especially for cross-environment settings. We hope that our dataset can first enable accurate HPE using these wireless sensors in the controlled condition, and then extend our dataset to more angles, different locations, and even multiple subjects in the MM-Fi 2.0.
>
> To demonstrate if more complicated movements lead to more difficulties for HPE tasks, we add the following results (in the next rebuttal box due to the character limit) for each action in the S1 setting. As shown in the table below, the results of all actions are listed for more detailed comparisons. From the comparison between A15/A16 (lunge towards left/right direction) and A24/A25 (body extension towards left/right direction), we can observe that the errors of A15/A16 are slightly larger than those of A24/A25, although they all contain arm and leg actions toward the same direction, which indicates that the global movement would degrade the performance to some extent, especially for the WiFi modality. When we compare A15/A16 with A09/A10(lunge towards the left-front/right-front direction), the errors of A09/A10 are obviously larger than those of A15/A16, though these four lunge actions all contain the whole body movements, showing that the facing angle relative to the sensing platform will also influence the estimation errors. Therefore, we think more complicated actions such as whole body movement and various facing angles are more difficult tasks. We plan to extend these more complicated actions in the future.

---

> ### Author Response · Authors · 2023-08-17
> **Response to Reviewer boR4 (part 2)**
>
> | Modality  | RGB   | RGB      | LiDAR | LiDAR    | mmWave | mmWave   | WiFi   | WiFi     |
> |-----------|-------|----------|-------|----------|--------|----------|--------|----------|
> | Criterion | MPJPE | PA-MPJPE | MPJPE | PA-MPJPE | MPJPE  | PA-MPJPE | MPJPE  | PA-MPJPE |
> | A01       | 82.5  | 31.2     |  91.3 |   60.0   |  151.1 |   65.0   | 179.4  | 102.1    |
> | A02       | 54.8  | 28.6     |  75.3 |   60.9   |  82.2  |   40.9   | 154.5  | 119.4    |
> | A03       | 56.1  | 37.8     |  81.4 |   62.4   |  85.8  |   49.7   | 159.6  | 141.1    |
> | A04       | 64.0  | 35.7     | 105.2 |   68.7   |  86.7  |   52.2   | 165.6  | 103.9    |
> | A05       | 57.8  | 36.1     | 102.6 |   67.8   |  87.3  |   48.5   | 166.3  | 108.7    |
> | A06       | 65.0  | 42.0     |  98.0 |   76.1   |  79.4  |   47.1   | 234.9  | 121.5    |
> | A07       | 40.7  | 23.1     |  64.8 |   46.4   |  132.4 |   48.6   | 145.0  | 101.2    |
> | A08       | 39.8  | 25.4     |  69.0 |   50.4   |  193.3 |   51.6   | 135.6  | 103.1    |
> | A09       | 98.6  | 39.9     | 111.3 |   88.8   |  116.8 |   65.6   | 263.4  | 157.6    |
> | A10       | 72.4  | 36.7     |  98.7 |   75.3   |  117.6 |   63.2   | 244.7  | 152.0    |
> | A11       | 43.5  | 19.2     |  67.2 |   45.9   |  136.2 |   48.7   | 159.8  | 112.6    |
> | A12       | 58.7  | 31.6     |  85.5 |   60.4   |  102.3 |   55.1   | 211.4  | 120.5    |
> | A13       | 36.1  | 22.6     |  68.2 |   47.0   |  135.4 |   57.2   | 151.0  | 115.5    |
> | A14       | 49.6  | 31.6     |  71.1 |   48.5   |  139.3 |   51.2   | 138.0  | 115.8    |
> | A15       | 57.3  | 27.3     |  83.4 |   50.2   |  102.1 |   49.9   | 256.7  | 131.0    |
> | A16       | 58.0  | 26.3     |  87.3 |   50.7   |  100.9 |   46.8   | 252.9  | 132.0    |
> | A17       | 52.0  | 28.4     |  67.5 |   48.0   |  206.6 |   73.1   | 177.6  | 142.2    |
> | A18       | 48.6  | 28.8     |  72.1 |   50.8   |  265.8 |   76.5   | 165.5  | 146.7    |
> | A19       | 83.7  | 51.2     | 112.8 |   76.6   |  113.7 |   65.4   | 281.0  | 135.6    |
> | A20       | 71.2  | 40.9     | 100.7 |   69.5   |  96.2  |   59.9   | 203.9  | 120.7    |
> | A21       | 62.9  | 40.6     | 101.1 |   67.4   |  100.0 |   58.7   | 212.0  | 119.5    |
> | A22       | 73.3  | 38.8     |  92.7 |   64.2   |  98.3  |   51.3   | 199.6  | 112.0    |
> | A23       | 76.5  | 42.2     |  99.1 |   67.6   |  98.7  |   54.0   | 205.8  | 108.0    |
> | A24       | 48.5  | 25.9     |  80.2 |   47.2   |  139.1 |   47.9   | 170.3  | 100.8    |
> | A25       | 41.3  | 26.6     |  81.0 |   49.8   |  114.6 |   49.9   | 173.1  | 100.0    |
> | A26       | 88.2  | 30.8     |  94.8 |   61.9   |  93.1  |   55.3   | 307.5  | 156.2    |
> | A27       | 64.5  | 32.6     |  89.9 |   51.7   |  80.0  |   38.9   | 192.1  | 106.4    |

---

> ### Author Response · Authors · 2023-08-17
> **Response to Reviewer boR4 (part 3)**
>
> > Q3. ‘Resolution’ in Table 2 arises confusions. For example, mmWave has superior resolutions in depth, while maintaining resolutions in lateral dimensions. Does this ‘Resolution’ represent angular resolutions? Does ‘WiFi’ have generally poor resolutions in all dimensions?
>
> **Answer**: We apologize for the confusing word, and have revised the “Resolution” here to “Granularity”. Here we actually refer to the granularity of sensing data, i.e., how much information can be captured by a specific sensor. For example, the LiDAR captures dense point clouds while mmWave radar only captures very sparse point clouds, so the granularity of LiDAR is better than mmWave radar.
>
> > Q4. Considering the poor performance of WiFi, it is doubt that WiFi would provide additional gain when fused with other modalities.
>
> **Answer**: As shown in Table 3 and Table, we find that the results of R+W (radar+WiFi) are better than the results of the single radar. However, the improvement is very marginal considering the poor performance of WiFi. In real-world applications, the fusion weights will be very difficult to choose. Therefore, how to fuse multiple modalities in an automatic manner also deserves to be explored in the future, and our MM-Fi dataset can be utilized for evaluation.
>
> > Q5. S2, S3 demonstrated in L239—L240 and those explained in Section 5.2 are different.
>
> **Answer**: Thanks for pointing out this issue. We wrote this subsection in the wrong order. We have revised the manuscript accordingly.
>
> > Q6. It would be beneficial for providing detailed specifications of each sensor.
>
> **Answer**: We appreciate the suggestion and have added the detailed specifications of each sensor into the appendix. The supplementary contents are as follows:
>
> WiFi: We develop a customized OpenWrt firmware for COTS WiFi devices (TP-Link N750) using the Atheros CSI tool to enable a large-scale implementation of various CSI-enabled applications. Our platform reports all 114 subcarriers for the 40 MHz bandwidth on each antenna pair operating on 5 GHz. The platform has 3 pairs of antennas with one on transmitter and three on receivers to collect the CSI data based on our developed firmware tool, which finally provides a CSI data stream of 100Hz after average sliding on the raw data.
>
> Lidar: Ouster OS1 32-channel LiDAR is used to acquire dense point cloud data. It contains 32 vertical beams, which provides $\pm$0.7-5cm vertical angular resolution. Its vertical field of view is 45 degrees and its range is 120m. It can capture dense point clouds with a maximum of 1,310,720 points per second. In MM-Fi, we collected the raw data under 10 Hz frequency.
>
> mmWave radar: The Texas Instruments (TI) IWR6843AOP mmWave radar is used to collect mmWave point clouds with up to 30FPS. It is an integrated single-chip mmWave sensor that has 3 antennas to transmit FMCW and uses 4 antennas to receive reflected FMCW by utilizing a 60-64 GHz radio band. Under the condition of a 1.0-V internal LDO bypass model and 48% duty cycle, its typical power consumption is 1.75 W.
>
> RGB: The Intel RealSense camera D435, consisting of a depth module, an RGB module, and a Realsense vision processor D4, is a stereo solution, offering quality depth for a variety of applications. The RGB module applies the rolling shutter technology, enabling the maximal 1920x1080 high-resolution RGB frame acquisition at a sensing rate of 30 fps. Its horizontal and vertical fields of view are 69 degrees and 42 degrees, respectively. The depth module, formed with one IR projector and two imagers, could obtain a larger sensing field of view with 87 (horizontal) and 58 (vertical) degrees, and provide a 1280x720 resolution depth frame at up to 90 fps. What’s more, due to the global shutter technology, the depth module could even work in low-light situations, making the Realsense D435 a good solution for all-day depth sensing applications.

---

> ### Author Response · Authors · 2023-08-23
> **Looking forward to your reply**
>
> Dear Reviewer,
>
> We have answered the question and submitted the revised manuscript as well as supplementary materials. We are looking forward to your reply.
>
> Feel free to let us know if you have any other concerns. Thanks!
>
> Best Regards,
>
> Authors of Submission 44

---

> > ### Comment · Reviewer_boR4 · 2023-08-29
> >
> > Dear Authors,
> >
> > I feel appreciate that my concerns and suggestions have been well addressed in the modified version.
> >
> > However, I'm still feeling doubt that the data from mmWave radar and WiFi can be generalized with long-range, diverse facing angles, and moving conditions. Considering the low quality of radar point clouds and WiFi data, there is possibility of classifying the classes first, and then generate the memorized corresponding skeleton models.
> >
> > Thanks again for your hardwork in rebuttal.

---

> > > ### Author Response · Authors · 2023-08-29
> > >
> > > Dear Reviewer boR4,
> > >
> > > Thanks for the constructive suggestions that render the revised manuscript better.
> > >
> > > For sparse point clouds from mmWave radar, a potential solution is to do the person detection first, and then estimate the human pose from the point cloud inside the detection bounding box. For WiFi, the current data only supports the HPE task in a constrained environment, which aims to facilitate sports rehabilitation and somatosensory game applications. We will include your suggestions and extend the scenarios with more distances, facing angles, and moving conditions in the next MM-Fi 2.0 that contributes to ubiquitous HPE in any environment.
> > >
> > > Thanks again for your review and advice!
> > >
> > > Best regards,
> > > Authors of Submission 44

---

### Official Review · Reviewer_RJAA · 2023-07-22
**Review and Suggestions for Improvement on MM-Fi**

**Rating:** 7
**Confidence:** 4

**Strengths:**

1. Compared to published dataset on HPE, the MM-Fi is the first 3D HPE dataset with most of the non-intrusive sensing modalities,
including RGB, depth, LiDAR, mmWave, and WiFi, which addresses the issues of privacy intrusion and inconvenience associated with existing solutions that mainly rely on cameras and wearable devices for 4D human perception.

2. The dataset is annotated with various labels, which makes it suitable for various applications.

3.  The sensor platform used in the MM-Fi dataset is a valuable contribution to the field of multi-modal human perception, as it supports the collection and the synchronization of  multi-modal data.

**Additional Feedback:**

The structure of the article still has room for improvement. A major advantage of the non-intrusive multi-modal human dataset is privacy protection. Future work can consider how to complete downstream tasks while protecting user privacy. At the same time, the fusion method of multiple modalities in this paper is also worthy of further exploration.

**Clarity:**

The paper is well-written and provides detailed information on the dataset, making it a valuable resource for researchers in the field.

**Correctness:**

The dataset is documented and organized in a standard manner. The benchmarks were evaluated using standard metrics.

**Documentation:**

Regarding the dataset, the paper provides details on data collection and organization, including hardware composition, data modality, activity categories, participant composition. The paper also mentions the availability and maintenance of the dataset, providing a URL for reviewers to access the dataset and a GitHub repository for hosting, licensing, and maintenance. The paper also provides a toolbox for the dataset, improving the usability of the dataset. Regarding the benchmarks, the paper provides enough detail to support reproducibility, including information on training details, data splits, and hyperparameters used in the experiments.

**Ethics:**

No concern.

**Limitations:**

The authors have addressed the ethical considerations of their work and discussed the limitations in the appendix.

**Opportunities For Improvement:**

My first concern is the accuracy of data annotation. I look forward to the author explaining why a full-body motion capture system, such as OptiTrack, is not used to obtain precise spatial coordinates of 3D human key points. Whether the annotation using deep learning algorithms will lose accuracy and reduce the quality of the dataset?

In Section 5.1, the author listed the baseline methods, and a richer discussion is needed here. I would like the authors to explain why they chose these methods, and whether there are other competing algorithms or SOTAs that need to be experimented with.

Now that the authors emphasize that this dataset opens up new research possibilities, a wider variety of downstream tasks as well as more advanced multimodal fusion methods should be discussed and experimented with.

**Relation To Prior Work:**

Yes, this paper discussed the related work in Section 1 and 2, and demonstrated the difference from the prior work.

**Summary And Contributions:**

The paper introduces MM-Fi, the first multi-modal, non-intrusive 4D human dataset, encompassing 25 action categories with over 320,000 synchronized frames from five modalities, collected from 40 human subjects. This dataset is enhanced with annotations designed for tasks like human pose estimation and action recognition. Through detailed experiments, the authors evaluate the sensing potential of these modalities, positioning MM-Fi as a pivotal resource for advancing research in areas such as action recognition, human pose estimation, and interdisciplinary healthcare.

---

> ### Author Response · Authors · 2023-08-17
> **Response to Reviewer RJAA (1/2)**
>
> We sincerely thank the reviewer RJAA for the insightful and constructive comments. We are glad that the reviewer acknowledges that the dataset and the sensor platform are novel and can facilitate various tasks and applications. Here we answer all the questions and hope they can address the concerns.
>
> >Q1. The accuracy of data annotation. Why not use a full-body motion capture system? Will the annotation using deep learning algorithms lose accuracy and reduce the quality of the dataset?
>
> **Answer**: We did not use full-body motion capture (Mocap) systems due to several reasons. Firstly, one of the crucial features of MM-Fi is to provide data from multiple environments for cross-scene wireless human pose estimation, but the deployment of a Mocap system requires the devices to be installed at relatively fixed locations. Moving the entire equipment is very inconvenient, e.g., OptiTrack and Vicon. Secondly, these systems are of high cost, while our low-cost annotation solution also provides accurate annotations validated by manual annotation.
> In our work, we not only use deep learning algorithms but also take the triangulation and optimization in Eq(2) and Eq(3) into consideration, as well as the specific refinement for actions containing occlusion cases. We validate the quality of the data annotation by comparing a random subset with manual annotation in section 4.3. The reprojection error PCKh@0.5 is 95.66%, demonstrating the reliability of our annotation. We notice that such an annotation strategy is also applied to other dataset annotations [1][2]. In the future, we plan to extend the dataset to multi-user scenarios. To guarantee the quality, we will install the Mocap system in the MM-Fi 2.0.
>
> > Q2. The authors need a richer discussion on the baseline methods and explain the reasons for the choices.
>
> **Answer**: Thanks for the valuable advice. We have supplemented more details on why we chose these baseline methods. These have been added to the manuscript (appendix) for better clarification.
>
> WiFi: We utilize MetaFi++ as our baseline, which is based on a convolutional network and employs a self-attention mechanism (in transformer) to learn the importance of different antennas. This enables the selective fusion of relevant information from antennas, resulting in enhanced spatial capture capability and achieving state-of-the-art scores. There are approaches in the field of WiFi pose estimation, e.g., GoPose, WiPose, and Winect, all of which only use convolutional methods. Moreover, these methods handle WiFi data with only 30 subcarriers, while our MM-Fi has a higher granularity of data with 114 subcarriers, so the inconsistency of data prevents us from using these methods as baselines.
>
> LiDAR and mmWave radar: The current human pose estimation (HPE) methods for point cloud consist of convolutional networks, graph neural networks, and the recent transformer. We chose Point Transformer [1] as our point cloud backbone because it shows state-of-the-art performance on many point cloud recognition tasks. Its design utilizes the self-attention mechanism to minimize the computational complexity for each layer and maximize the number of parallelizable calculations, which perfectly matches the characteristic of point cloud data.
>
> RGB: The visual baseline (VideoPose3D) is a popular 2D-to-3D solution that transforms 2D keypoints into 3D joints, developed by Facebook Research. We choose this baseline since it has been widely utilized in many human pose estimation benchmarks [1][2]. There are many other works in computer vision for 3D human pose estimation, and we welcome the community to supplement more benchmarking results on our dataset.

---

> ### Author Response · Authors · 2023-08-17
> **Response to Reviewer RJAA (2/2)**
>
> > Q3: A wider variety of downstream tasks as well as more advanced multimodal fusion methods should be discussed.
>
> **Answer**: MM-Fi significantly expands the horizons of research by furnishing meticulously synchronized multimodal human sensing data. As advised by the reviewer, we have encapsulated these novel research directions as follows, and have incorporated these enhancements into the manuscript:
>
> Cross-domain wireless sensing. The realm of wireless sensing is often challenged by recognition performance disparities stemming from domain shifts, which arise due to variations in environments and subjects. While this issue has garnered extensive exploration within action recognition based on RGB data, it remains relatively uncharted within the context of Human Pose Estimation (HPE) utilizing modalities such as mmWave, LiDAR, and WiFi. The MM-Fi dataset presents a unique opportunity to address this gap by facilitating research into domain adaptation and generalization for HPE on these emerging modalities. By enabling investigations into the adaptation of models across diverse domains, MM-Fi opens avenues for enhancing the robustness and applicability of wireless sensing technologies.
>
> Cross-modal supervision for fine-grained wireless sensing. Previous works have showcased the utility of WiFi and mmWave radar for action recognition tasks. However, MM-Fi introduces a transformative dimension by offering meticulously synchronized multimodal data and comprehensive annotations. Based on MM-Fi, cross-modal learning can enable these sparse data modalities to achieve fine-grained recognition tasks, e.g., human pose estimation and action segmentation.
>
> Multi-modal wireless sensing. MM-Fi's integration of five distinct modalities empowers researchers to explore the potential of multi-modal wireless sensing, where different sensing technologies complement each other to achieve more comprehensive and accurate insights into human behavior. For example, integrating RGB and WiFi can overcome the illumination issue of RGB-based solutions. This dataset serves as a launchpad for pioneering investigations into techniques that fuse information from LiDAR, mmWave radar, and WiFi signals to attain a holistic understanding of human actions and interactions. As researchers delve into multi-modal fusion methods, the MM-Fi dataset becomes a valuable resource for the development of advanced solutions that leverage the strengths of each modality, while compensating for their individual limitations.
>
> >Q4: A major advantage of the non-intrusive multi-modal human dataset is privacy protection. Future work can consider how to complete downstream tasks while protecting user privacy. At the same time, the fusion method of multiple modalities in this paper is also worthy of further exploration.
>
> **Answer**: We appreciate the suggestions from the reviewers. Privacy-preserving human sensing will play a crucial role in hospitals and smart homes. The MM-Fi 2.0 will be designed to include more privacy-preserving scenarios and human activities.
>
> >Q5: The authors provide the limitations of work in terms of annotation and fusion benchmark. In my opinion, there are also limitations in terms of movement diversity (i.e., each subject is fixed in a certain location and facing angle).
>
> **Answer**: We agree with the reviewers and have added these limitations to our limitation subsection. Thanks!
>
> **References**
>
> [1]. Sizhe An, Yin Li, and Umit Ogras. mRI: Multi-modal 3d human pose estimation dataset using mmwave, RGB-d, and inertial sensors. In Thirty-sixth Conference on Neural Information Processing Systems Datasets and Benchmarks Track, 2022.
>
> [2]. Cai, Z., Ren, D., Zeng, A., Lin, Z., Yu, T., Wang, W., ... & Liu, Z. (2022, October). Humman: Multi-modal 4d human dataset for versatile sensing and modeling. In European Conference on Computer Vision (pp. 557-577). Cham: Springer Nature Switzerland.
>
> [3] Zhao, H., Jiang, L., Jia, J., Torr, P. H., & Koltun, V. (2021). Point transformer. In Proceedings of the IEEE/CVF international conference on computer vision (pp. 16259-16268).

---

> ### Author Response · Authors · 2023-08-18
> **Looking forward to your reply**
>
> Dear Reviewer,
>
> We have answered the question and submitted the revised manuscript as well as supplementary materials. We are looking forward to your reply.
>
> Feel free to let us know if you have any other concerns. Thanks!
>
> Best Regards,
>
> Authors of Submission 44

---

> > ### Comment · Reviewer_RJAA · 2023-08-30
> >
> > Dear authors,
> >
> > I appreciate your responses and effort on the paper revision. Your responses and modifications have addressed most of my concerns and questions.
> >
> > Regarding the specific points from your responses:
> >
> > **Regarding Q5**: I still believe that the lack of movement diversity in the dataset remains a major limitation of this work. I look forward to seeing these issues addressed and improved in MM-Fi 2.0.
> >
> > **Regarding Q2**: I appreciate the more detailed discussion and clarification provided by the authors. However, I believe there are more SOTAs (state-of-the-art models) that need to be considered and compared. For instance, some pre-trained pose estimation models, such as MotionBERT and PointGPT, have outperformed the VideoPose3D and PointTransformer methods mentioned by the authors across multiple datasets. I look forward to seeing more baseline results on this dataset in the authors' future work.
> >
> > Given your explanations and my concerns, I have decided to maintain my original score.
> >
> > Best,
> >
> > Reviewer RJAA

---

> > > ### Author Response · Authors · 2023-08-31
> > >
> > > Dear Reviewer RJAA,
> > >
> > > We appreciate your responses and constructive suggestions.
> > >
> > > Regarding the responses:
> > >
> > > **Regarding Q5**: We agree that the current dataset lacks movement diversity. The current dataset is collected to facilitate wireless HPE tasks under the constrained environment, which enables various applications including rehabilitation and motion sensing games based on radar, LiDAR and WiFi. We have written the limitations and future plan in the appendix. We will definitely include movement diversity in the future MM-Fi 2.0, which can help improve the model generalization ability.
> > >
> > > **Regarding Q2**: As the reviewer points out, for RGB-based HPE, we notice that new SOTA methods have been proposed (e.g., MotionBERT and PointGPT). We hope that MM-Fi also contributes to building GPT or transformer-based models in wireless sensing fields using these IoT sensors. More baseline results will be compared in future work.
> > >
> > > Thanks for the response!
> > >
> > > Best Regards,
> > >
> > > Authors of Submission 44

---

### Author Response · Authors · 2023-08-17
**Summary of the rebuttal and the revised manuscript**

> We sincerely thank all the reviewers for their helpful comments and suggestions. We now have uploaded the rebuttal version of our paper together with the appendix where the revisions are marked in magenta. Due to the space limit, we put the discussions, additional experiments, and more details about the sensor platform in the appendix. We hope the revamped manuscript can be satisfactory.

Here is the summary of the major changes we made in the revision:

1. We include more references on multimodal datasets and make the comparison in Table 1.
2. We include details on sensor specifications and collection environments.
3. We include the details on the baseline models and the reason why we choose them.
4. We include one more annotation to segment the action.
5. We retrain the RGB model from scratch and add the results.
6. We add discussions on future work and research enabled by MM-Fi.
7. We fix some typos and writing issues in the manuscript.

---

### Decision · Program_Chairs · 2023-09-22

**Decision:**

Accept (Poster)

**Comment:**

This paper got 7, 6, 6, 6, 7 (the last reviewer raised the score from 5 to 7).

The reviewers express a general consensus on the novelty and importance of the MM-Fi dataset paper, which introduces an innovative multi-modal human sensing dataset. Its real-world application, with a combination of multiple sensing modalities such as RGB cameras, depth cameras, LiDAR, mmWave, and WiFi, is acknowledged as pioneering in the field. The comprehensive explanation of the data collection process, including signal gathering, synchronization, and annotation, has been appreciated across the board. The analysis and data visualization provided are noted as robust and clear.

However, there are some shared concerns: a limited number of sequences in the dataset, more granularity in the actions, and the lack of representative daily activities. There's a significant concern raised about the clarity and motivation behind using certain modalities like WiFi, given the performance drop observed in human pose estimation. Missing references and comparison discussion, especially the mRI dataset, which is required to be clearly discussed to clarify the novelty. A few technical inconsistencies and suggestions for clearer presentation have been pointed out that need to be addressed for clarity.

The authors did a good job with the rebuttal. All the reviewers acknowledged the rebuttal. Given the rebuttal and revision, this AC shares the viewpoints of the reviewers by its own merits and considerable effort in this paper.